# EAGLES: Towards Effective, Efficient, and Economical Federated Graph Learning via Unified Sparsification

Zitong Shi [* 1]   Guancheng Wan [* 1]   Wenke Huang [* 1]   Guibin Zhang [2]   He Li [1]   Carl Yang [3]   Mang Ye [1]

## Abstract

Federated Graph Learning (FGL) has gained significant attention as a privacy-preserving approach to collaborative learning, but the computational demands increase substantially as datasets grow and Graph Neural Network (GNN) layers deepen. To address these challenges, we propose **EAGLES**, a unified sparsification framework. EAGLES applies client-consensus parameter sparsification to generate multiple unbiased subnetworks at varying sparsity levels, reducing the need for iterative adjustments and mitigating performance degradation. In the graph structure domain, we introduced a dual-expert approach: a *graph sparsification expert* uses multi-criteria node-level sparsification, and a *graph synergy expert* integrates contextual node information to produce optimal sparse subgraphs. Furthermore, the framework introduces a novel distance metric that leverages node contextual information to measure structural similarity among clients, fostering effective knowledge sharing. We also introduce the **Harmony Sparsification Principle**, EAGLES balances model performance with lightweight graph and model structures. Extensive experiments demonstrate its superiority, achieving competitive performance on various datasets, such as reducing training FLOPS by 82% ↓ and communication costs by 80% ↓ on the ogbn-proteins dataset, while maintaining high performance. The code is anonymously available at this link.

## 1. Introduction

Federated Graph Learning (FGL) has emerged as a pivotal field decentralized machine learning, harnessing the

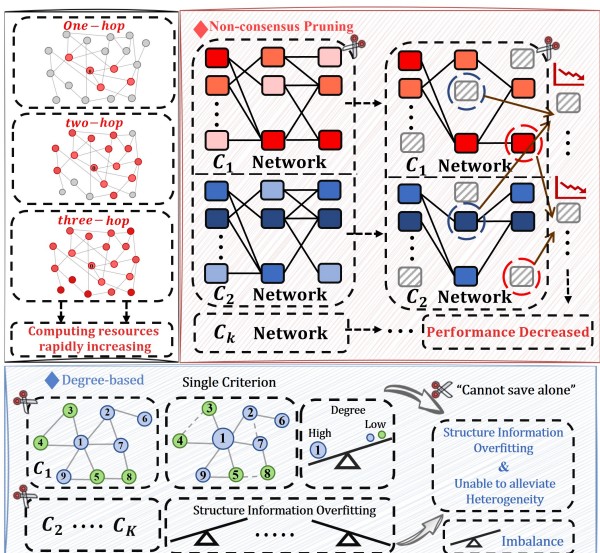

Figure 1: **Problem Illustration.** The message-passing mechanism in GNNs leads to a significant increase in computational demand as dataset sizes grow and the number of GNN layers deepens. Existing studies attempt to reduce computational requirements through model pruning or graph sparsification. However, model pruning approaches often overlook the variability in parameter importance across different clients, while certain graph sparsification techniques rely on a single pruning criterion and fail to account for data heterogeneity, thus limiting their effectiveness.

combined strengths of Federated Learning (FL) systems (Li et al., 2020; Yang et al., 2016; 2019; Mammen, 2021) and Graph Neural Networks (GNNs) (Luo & Wu, 2022; Lee et al., 2019; Gao & Ji, 2019; Wan et al., 2024b). This approach allows multiple clients to collaboratively train without sharing their raw data, ensuring data privacy while effectively extracting knowledge from graph-structured information. It has already been widely applied in various domains such as social networks (Goldenberg, 2021; Barnes, 1969), healthcare (Schrodt et al., 2020; Li et al., 2022b), finance (Cardoso et al., 2020; Saha et al., 2022), and recommendation systems (Gao et al., 2023; Wang et al., 2021; Ma et al., 2025). However, as graph data scales continue to expand, the computational cost of message passing grows rapidly, posing significant challenges to the computational resource demands during the training phase of GNNs.

To address this inefficiency and obtain a better-performing

---

[*]Equal contribution  [1]National Engineering Research Center for Multimedia Software, School of Computer Science, Wuhan University, Wuhan, China [2]National University of Singapore, Singapore [3]Department of Computer Science, Emory University, USA. Correspondence to: Mang Ye <yemang@whu.edu.cn>.

*Proceedings of the $42^{nd}$ International Conference on Machine Learning*, Vancouver, Canada. PMLR 267, 2025. Copyright 2025 by the author(s).

model, numerous effective methods have been proposed. These approaches can be mainly categorized into four types: graph simplification optimization (Chen et al., 2021b;a; Sohrabi et al., 2021) and model parameter tuning (Ma, 2024; Yi et al., 2024; Liang et al., 2024; Nguyen et al., 2024), graph-based knowledge distillation (Zhu et al., 2021; Wang et al., 2024b; Zhu et al., 2024), graph structure completion (Baek et al., 2023; Chen et al., 2022; Liang et al., 2024). The former two groups focus on simplifying graph structures and adjusting model parameters. Graph simplification optimization typically reduces the training burden through sampling or pruning. However these strategies often based on a single selection strategy (Li et al., 2022a; Parchas et al., 2018; Fung et al., 2011), which leads to the neglect of critical information inherent within the graph structure. We refer to this phenomenon as *structure information overfitting*. In model parameter tuning, some methods (Ma, 2024; Yi et al., 2024) consider lightweighting models by pruning redundant parameters. However, these approaches necessitate iterative adjustments to the pruning rate and are not fully suitable for federated systems, due to the fact that the importance of parameters may vary across different client models for the same layer. We present radar charts comparing advanced centralized methods: Dspar and ACE-GLT with our method, as shown in Figure 2. The results indicate that advanced methods relying solely on a single approach and disregarding federated characteristics also face a significant risk of model performance degradation. Based on the observations, we raise the following question: **I)** *How can we streamline models to reduce the training burden while preserving critical graph information?*

Graph-based knowledge distillation aggregates by transmitting logits or graph embeddings instead of model parameters. However, neither logits nor graph embeddings effectively capture structural information, and both introduce additional computational overhead. As regards the graph structure completion, certain methods endeavor to produce missing neighbors and their corresponding features for peripheral nodes, aiming to enhance the trainability of these damaged or low-quality nodes. However, these generative methods encounter even greater computational overhead compared to graph-based knowledge distillation, particularly in large-scale graphs. Although both methods aim to enhance model performance or mitigate heterogeneity, they all come with substantially higher computational costs, leading to a strong dependence on more hardware resources. This leads us to consider: **II)** *How can we effectively mitigate structural heterogeneity while minimizing additional computational demands?*

To simultaneously overcome the aforementioned questions, we introduce **EAGLES**: Towards **E**ffective, Efficient, **A**nd Economical Federated **G**raph **L**earning via Unifi**E**d **S**parsification. To address **I)**, we first introduce a unified

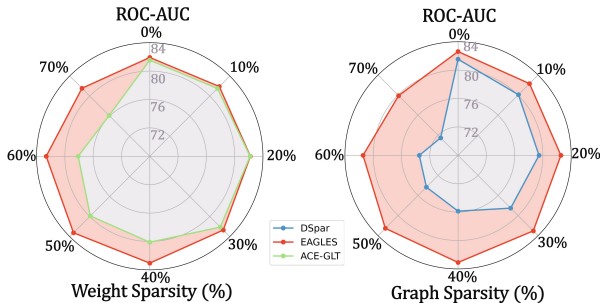

Figure 2: **ROC-AUC Comparison on Ogbn-Proteins using 4-layer DeeperGCN**: The radial axis represents ROC-AUC. *(left)* ACE-GLT focuses on parameter-level sparsification, while *(right)* Dspar specializes in graph-level sparsification. For a detailed explanation of the ROC-AUC metric, please refer to Appendix B.

sparsification approach, applied at both the graph level and the model parameter space. At graph level, we deploy two types of expert models at the client to solve *structure information overfitting*, referred to as sparsification experts and synergy experts. Each sparsification expert specializes in a particular sparsification criterion, such as jaccard similarity, spectral sparsification, and others. They perform sparsification on the one-hop subgraph of each node based on specific expertise. To better integrate the sparsified subgraphs from multiple experts, synergy expert models node features employing *hard concrete distribution* to learn structural information and dynamically select and integrate key information from different sparsified versions for each node. By the combined effect of the two types of expert, we reduce graph size, enabling more **efficient** message passing and *alleviating structure information overfitting*. While at the parameter space, we allow sub-networks with various pruning rates to share weight parameters and thus avoid the need for iterative training. This approach produces multiple sparsified networks in a single pass, reducing parameters and achieving a more **economical** communication overhead.

To tackle **II)**, we utilize the node contextual knowledge learned by the graph synergy expert, applying it to a novel transmission distance metric for the purpose of assess structural similarity among clients' data. This approach *encourages structurally similar clients to share more knowledge*. While parameter sparsification is performed, we consider the varying importance of parameters at corresponding positions across clients. To achieve this, we introduce a consensus-based parameter sparsification strategy. This strategy includes *layer-wise alignment of the personalized parameter sparsification masks*. With aggregation adjustments and parameter consensus sparsification, we more accurately account for unbiased knowledge sharing, **effectively** mitigating the performance degradation caused by heterogeneity. Additionally, we introduce the **Harmony Sparsification Principle** as the guiding principle for the above process. Thereby we establish an efficient, economical unified and effective sparsification framework. Our principal contribu-

tions are summarized as follows:

- We identify that existing sparsification methods lack a unified framework, considering only graph level or parameter level sparsification. Moreover, they fail to ensure effectiveness while achieving efficiency and economy.
- We are the first to introduce unified sparsification in FGL. This approach performs both graph and parameter space, minimizing computational and communication demands while maintaining or improving model performance.
- We conducted extensive experiments on both small and large graphs, evaluating our method across various metrics to confirm its superiority. For instance, under the Ogbn-Proteins + DeeperGCN setting, it achieves state-of-the-art (SOTA) performance while reducing training FLOPS by 82% ↓ and communication bytes by 80% ↓.

## 2. Related Work

### 2.1. Federated Graph Learning

Federated Graph Learning (FGL) combines the dual advantages of federated systems and Graph Neural Networks (GNNs) (Wu et al., 2020; Wan et al., 2025b; Shi et al., 2024; Wan et al., 2025a), enabling the training of graph-structured data while preserving data privacy (He et al., 2021; Jin et al., 2022; Wang et al., 2020; Wu et al., 2021; Zhang et al., 2022; Wan et al., 2024a; 2025c). While numerous studies have concentrated on improving model performance, as datasets continue to grow, reducing computational resource consumption have gradually become pressing challenges (Liu et al., 2023b; Chen et al., 2023a; 2024). Existing methods are primarily focused on sparsification within the parameter space of traditional FL (Zeng et al., 2022; Jiang et al., 2023; Fang et al., 2025). To the best of our knowledge, we are the first to propose a unified framework that simultaneously sparsifies both the graph structure and parameter space. More importantly, our approach successfully mitigates data heterogeneity and maintain or enhances model performance.

### 2.2. Mixture of Experts

The Mixture of Experts (MoE) has emerged as an effective approach in scaling deep learning models. By dynamically selecting and combining the outputs of specialized expert networks, MoE enables the model to perform more efficiently and accurately across different tasks or inputs. In graph learning, moE has also been widely applied, including large-scale graph learning (Wang et al., 2024a; Gupta et al., 2022; Jacobs et al., 1991), multi-task learning (Liu et al., 2023a; Chen et al., 2023b; Ma et al., 2018; Shazeer et al., 2017; Bi et al., 2025a;b;a), and graph classification (Hu et al., 2021; Yao et al., 2019). In our work, we leverage MoE to achieve multi-criteria sparsification at the graph level, alleviating data heterogeneity and enabling structurally similar clients to share more knowledge.

### 2.3. Graph Sparsification & Model Pruning

Graph sparsification and model pruning are widely studied techniques for reducing computational complexity while maintaining model performance. Graph sparsification aims to retain the most important structural components of a graph by removing less critical edges or nodes (Gong et al., 2021; Müller et al., 2022; Liu et al., 2023c; Batjargal et al., 2019; Wan & Schweitzer, 2021). On the other hand, model pruning focuses on removing redundant model parameters to create more lightweight architectures (Yi et al., 2024; Qiu et al., 2022; Wu et al., 2023a). For instance, DSpar selectively retains the connections of high-degree nodes to sparsify the graph structure (Liu et al., 2023c; Cai et al., 2024b). PR-FL employs an approach where model pruning is performed first, followed by the gradual recovery of model size during the training process (Ma, 2024; Cai et al., 2024a). However, as mentioned earlier, both methods face different limitations. We aim to overcome these shortcomings and obtain a better-performing lightweight model.

## 3. Preliminary

**Notations.** Following the general paradigm of federated graph learning, there are $K$ participants (indexed by $k$), each client $C_k$ possesses its own private data, represented as $\mathcal{G}_k = (\mathcal{V}_k, \mathcal{E}_k)$. Here, $\mathcal{V}_k = \{v_i\}_{i=1}^{N_k}$ denotes the set of nodes, containing $|\mathcal{V}_k| = N_k$ nodes, while $\mathcal{E}_k = \{e_{mn}\}_{m,n}$ represents the set of edges. The adjacency matrix for the $k$-th client's graph $\mathcal{G}_k$ is denoted as $\mathbf{A}_k = \{A_{ij}\}_{i,j}$, where $A_{ij} = 1$ if there is an edge between nodes $v_i$ and $v_j$, and $A_{ij} = 0$ otherwise. Additionally, each client has a feature matrix $\mathbf{X}_k$, where each row corresponds to the features of a specific node in $\mathcal{V}_k$. We define the global model at the beginning of the $t$-th communication round as $\mathcal{M}^t$, and the model of the $k$-th local client as $\mathcal{M}_k^t$ with parameters $\theta_k^t$.

### 3.1. Graph & Parameters Sparsification

For the graph sparsification task, we define the sparsified set of edges as $\mathcal{E}_k^s$, and the sparsified graph representation of the $k$-th client as $\mathcal{G}_k^s = \{\mathcal{V}_k, \mathcal{E}_k^s\}$. The one-hop subgraph of node $v_i$ is denoted as $\mathcal{G}_{k,v_i}^s$. We define the parameter matrix of the $k$-th local model's $l$-th layer as $\mathbf{W}_k^{(l)}$. Its sparsified version is denoted as $\mathbf{W}_k^{(l),s}$. The sparsification rates for the graph and weights $S^{\mathcal{G}}$ and $S^{\mathbf{W}}$ are defined as follows:

**Definition 1.** (Graph Sparsification Ratio). *For a sparsified graph $\mathcal{G}^s$, we calculate its corresponding graph sparsification rate as follows:*

$$\mathcal{G}^s = (\mathcal{V}, \mathcal{E}^s), \quad S^{\mathcal{G}} = \frac{|\widetilde{\mathcal{E}}|}{|\mathcal{E}|}. \tag{1}$$

**Definition 2.** (Model Sparsification Ratio). *For a sparsified model $\mathcal{M}$ with $L$ layers, we calculate its parameter sparsification rate as follows:*

$$\mathcal{M}^s = \{\mathbf{W}^{(1)}, \mathbf{W}^{(2)}, \dots, \mathbf{W}^{(L)}\}, S^{\mathbf{W}} = \frac{\sum_{l=1}^{L} |\widetilde{\mathbf{W}}^{(l)}|}{\sum_{l=1}^{L} |\mathbf{W}^{(l)}|}. \tag{2}$$

where $|\cdot|$ represents the cardinality (*i.e.*, the number of elements) of a set, and $\widetilde{\phantom{x}}$ denotes the sparsified portions. Our objective is to maintain or improve the performance of the local model while maximizing $S^{\mathcal{G}}$ and $S^{\mathbf{W}}$.

## 3.2. Sparsification Criterion Design

Current sparsification research in FGL lacks consideration for data structure heterogeneity and lacks a unified framework to simultaneously address both graph level and parameter space. These constraints prevent deep sparsification, as illustrated in Figure 2. We argue the conditions that an ideal sparsification architecture should satisfy as follows:

> **Ideal Sparsification Architecture for FGL**: *Given a sparsification task q, an optimal FGL architecture to achieve this should fulfill the following specifications:* **Effectiveness**: *The sparsification structure must effectively mitigate data heterogeneity.* **Completeness**: *The sparsification structure should comprehensively account for redundant components (i.e., graph and parameter space).* **Scalability**: *The sparsification architecture should be capable of efficient operation on large-scale datasets.*

To fulfill the specifications above, we introduce the Harmony Sparsification Principle to guide the criteria for designing sparsification architectures in FGL:

**Principle 1.** (Harmony Sparsification Principle). *Given a graph $\mathcal{G} = \{\mathcal{V}, \mathcal{E}\}$ and a L-layer model $\mathcal{M} = \{\mathbf{W}^{(1)}, \mathbf{W}^{(2)}, \ldots, \mathbf{W}^{(L)}\}$, along with two ideal sparsifiers $\mathcal{F}_g \in \{0,1\}^{|\mathcal{E}|}$ and $\mathcal{F}_p \in \{0,1\}^{|\mathcal{M}|}$, responsible for graph and parameter sparsification, respectively. $\mathcal{G}^s$ and $\mathcal{M}^s$ should satisfy the following condition:*

$$\underset{\mathcal{F}_g, \mathcal{F}_p}{\arg\min} E \left( \sum_{\mathcal{N}' \in \{\mathcal{N}\}} \mathcal{N}'(\mathcal{G}, \mathcal{G}^s) + \sum_{\mathcal{A}' \in \{\mathcal{A}\}} \mathcal{A}'(\mathcal{M}, \mathcal{M}^s) \right)$$
$$+ \underset{\mathcal{F}_g, \mathcal{F}_p}{\arg\max} E \left( \sum_{\mathcal{Q}' \in \{\mathcal{Q}\}} \mathcal{Q}'(\mathcal{G}, \mathcal{G}^s) + \sum_{\mathcal{Z}' \in \{\mathcal{Z}\}} \mathcal{Z}'(\mathcal{M}, \mathcal{M}^s) \right) \tag{3}$$

*where $E$ denotes the mathematical expectation, $\{\mathcal{N}\}$ represents metrics for evaluating differences in graph structure (e.g., Jaccard similarity, spectral sparsification, etc.), $\{\mathcal{A}\}$ represents metrics for assessing model parameters in terms of inference performance (e.g., Top-k Accuracy, ROC-AUC, etc.), $\{\mathcal{Q}\}$ evaluates metrics related to the graph message-passing load (e.g., node load, betweenness centrality, etc.), and $\{\mathcal{Z}\}$ includes metrics for evaluating reductions in computational resources (e.g., training flops, communication bytes, etc.).*

## 4. Methodology

### 4.1. Overview

The proposed EAGLES can be decomposed into two components: Consensus-Based Parameter Sparsification and

Heterogeneous-Aware Graph Sparsification, which correspond to section 4.2, section 4.3, respectively. We use dynamic rollback pruning for layer-wise parameter sparsification, performing row-wise pruning within each layer to obtain subnetworks at different sparsity levels. Based on task requirements, we select and retrain suitable subnetworks. Subsequently, we leverage sparsification and synergy experts to remove redundant edges and unify client structures. Ultimately, we upload both synergy expert and backbone parameters, using Optimal Transport to assign higher weights to clients with similar structures, allowing the local model to absorb more knowledge from similar data distributions. The framework of the method is shown in Figure 3.

### 4.2. Consensus-Informed Parameter Sparsification

**Motivation.** GNNs in both centralized and federated distributed environments face significant computational demands and extended training times, particularly on large datasets (Hamilton et al., 2017; Chen et al., 2017). However, traditional model sparsification approaches rely on multiple iterative cycles with preset pruning rates, directly conflicting with the aim of reducing computational load. Moreover, existing pruning methods in FL fail to effectively consider unbiased knowledge sharing, leading to notable efficiency losses.

**Dynamic Mask Thresholds.** Traditional lottery ticket networks require manually setting sparsity levels, selecting a fixed proportion of parameters with the smallest absolute values for pruning. We define the threshold vector for the $l$-th layer as $\boldsymbol{\kappa}_\theta^l$, corresponding to the parameter matrix $\mathbf{W}^l$ in this layer. The parameter mask $\boldsymbol{m}_{ij}^{(l)}$ for the entry located at the $i$-th row and $j$-th column of $\mathbf{W}^l$ is computed as follows:

$$\boldsymbol{m}_{ij}^{(l)} = \begin{cases} 1, & \text{if } |\mathbf{W}_{ij}^{(l)}| \geq \boldsymbol{\kappa}_{\theta,i}^{(l)}, \\ 0, & \text{otherwise}, \end{cases} \tag{4}$$

where $\mathbf{W}_{ij}^{(l)}$ denotes the weight in the $i$-th row and $j$-th column, and $\boldsymbol{\kappa}_{\theta,i}^{(l)}$ denotes the $i$-th element of the vector $\boldsymbol{\kappa}_\theta^{(l)}$. **Dynamic threshold optimization.** In the forward pass, we define a simple binary step function $BS(x)$ to generate a binary mask based on whether the input exceeds a threshold. However, this discrete value is non-differentiable during the backward pass. To address this and facilitate effective threshold tuning, we employ the straight-through estimator (STE) method for optimization, whose effectiveness has been demonstrated in prior works (Bengio et al., 2013; Jang et al., 2016; Huh et al., 2023; Wu et al., 2023b; Yin et al., 2019). Specifically, in the backward pass, we approximate the gradient of $BS$ as the gradient of an identity function:

$$\frac{d\,BS(x)}{dx} = \begin{cases} 0, & \text{if } 0 \leq |x| < 0.5. \\ 1, & \text{otherwise.} \end{cases} \tag{5}$$

This approach allows gradients to bypass the non-

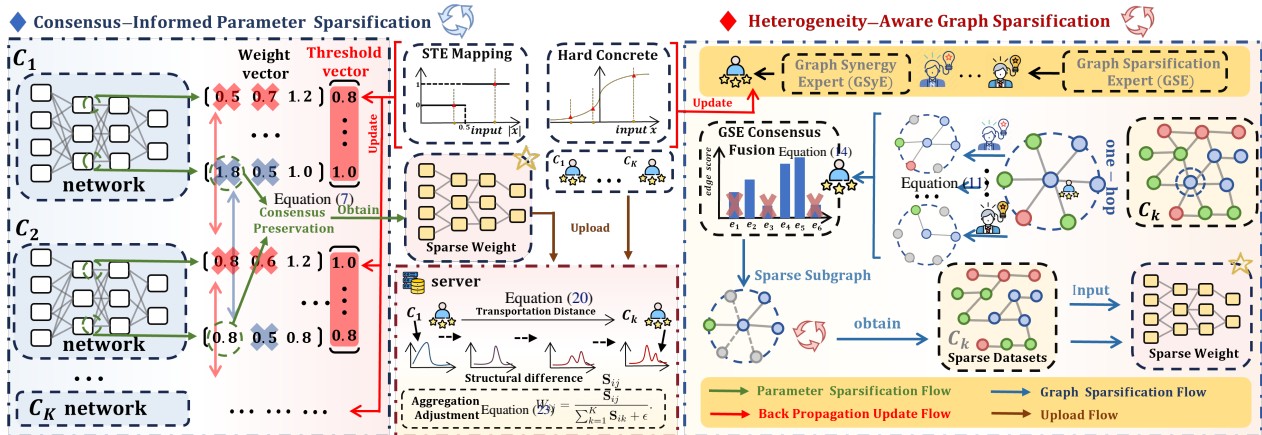

Figure 3: Architecture illustration of EAGLES. *(Left)* Consensus-Informed Parameter Sparsification is depicted, where the sparsification threshold vector is optimized using the Straight-Through Estimator (STE). *(Right)* Heterogeneity-aware graph sparsification is shown, with the Graph Synergy Expert optimized in reverse via the Hard Concrete distribution. *(Middle)* The server adjusts aggregation weights according to structural differences. Best viewed in color. Zoom in for details.

differentiability of *BS* during the backward pass, enabling effective updates to the threshold parameter $\kappa_\theta^{(l)}$. The loss function for this part is simply defined as follows:

$$\mathcal{L}_{Para} = \sum_l \exp(-\kappa_\theta^{(l)}). \tag{6}$$

**Consensus Parameter Mask Alignment.** Client-specific sparsification masks can lead to progressive overfitting to local data, a challenge that becomes even more pronounced during model pruning. Building on this, we propose a client-consensus-based parameter pruning scheme. After applying dynamic masking on each client, the resulting mask vector (in boolean form) is compressed into bit storage, with each 8-bit segment occupying a single byte. This approach significantly reduces communication overhead while enabling efficient selection of consensus parameters.

$$\left.\begin{array}{l} C_1 \ [1,0,1,0,\cdots,1,1,0] \\ \vdots \quad \Updownarrow \ \Updownarrow \quad \Updownarrow \\ C_k \ [1,1,1,0,\cdots,1,0,0] \\ \vdots \quad \Updownarrow \quad \Updownarrow \\ C_K \ [1,1,1,1,\cdots,1,0,0] \end{array}\right\} \rightarrow [1,1,1,1,\cdots,1,1,0]$$

**Final Mask Vector** (7)

This approach promotes pruning stability at a macro level while preserving a degree of generalization in the local models. To further reinforce deep pruning, we employ a dynamic rollback strategy. Specifically, we designate the highest-performing subnetwork within an accuracy range of ±3% around each multiple of ten as the optimal subnetwork for that pruning point. Before progressing to the next pruning point, we roll back to the optimal subnetwork from the previous point, ensuring stable and effective pruning depth.

### 4.3. Heterogeneity-Aware Graph Sparsification

**Motivation.** The information aggregation mechanism of GNNs imposes substantial computational demands, with time complexity growing explosively as the depth of aggregation increases (Chen et al., 2017; 2018; Chiang et al., 2019; Rong et al., 2019; Velickovic et al., 2019). There is limited research on sparsification within federated distributed settings. Clients in FGL often exhibit substantial structural differences, and poorly configured pruning methods can intensify this structural heterogeneity. Moreover, most existing sparsification methods rely on a single criterion. This reliance tends to overly preserve specific structural information aligned with that criterion, overlooking other aspects and resulting in *structure information overfitting*.

**Graph Sparsification Expert.** The Graph Sparsification Expert (GSE) is the central module for graph sparsification. Each GSE applies a unique sparsification criterion, evaluating each edge within the one-hop subgraph of each node according to its specific scoring standard. Specifically, we concatenate the feature matrix $\mathbf{X}_i$ of node $v_i$, the feature matrix $\mathbf{X}_j$ of its neighboring node $v_j$, and the weight $|e_{ij}|$ of the edge to form the input feature matrix for the GSE:

$$\mathbf{X}_i, \mathbf{X}_j \in R^{N \times D}, \qquad |e_{ij}| \in R^{1 \times 1},$$
$$\mathcal{X}_{ij}^{GSE} = [\mathbf{X}_i, \mathbf{X}_j, |e_{ij}| \cdot \mathbf{1}_{N \times D}] \in R^{N \times (2D+1)}. \tag{8}$$

Here, $|e_{ij}|$ is treated as a vector to enable broadcasting, and $\mathcal{X}_{ij}$ serves as the input matrix for the GSE to compute the edge score $o_j$ corresponding to $e_{ij}$. This process is repeated $N_{v_i}$ times, where $N_{v_i}$ denotes the number of neighbors of node $v_i$, ultimately yielding an ordered set of edge scores:

$$\mathbf{O}_{v_i} = \{o_1, o_2, \ldots, o_{N_{v_i}}\}. \tag{9}$$

The graph sparsity rate is a predefined hyperparameter, based on which we determine the final sparsified subgraph for this GSE:
$$\mathbf{O}_{v_i}^s = \{o_k, o_{k+1}, \ldots, o_{N_{v_i}}\}.$$
$$\mathcal{G}_{v_i}^s = \{\mathcal{V}, \mathcal{E}_{v_i}^s\}, \quad \mathcal{E}_{v_i}^s = \{e_k, e_{k+1}, \ldots, e_{N_{v_i}}\}. \tag{10}$$

Assuming that $t$ different GSEs have each produced a sparsified version for $v_i$, we obtain a set of sparsified graphs:

$$\mathbf{G}^s_{v_i} = \{\mathcal{G}^{s,katz.}_{v_i}, \mathcal{G}^{s,Jac.}_{v_i}, \ldots, \mathcal{G}^{s,Cos.}_{v_i}\}. \tag{11}$$

**Graph Synergy Expert.** To integrate the sparsification results from different experts, we introduce a Graph Synergy Expert (GSyE), driven by a gating mechanism (Greff et al., 2016; Li et al., 2015; Srivastava et al., 2015). However, the activation or deactivation of multiple sparsification experts poses a discrete binary challenge, making it non-differentiable in the backward pass. We propose optimizing the GSyE employing a hard concrete distribution to address this issue. The optimization of $\mathcal{G}^s_i$ is essentially an optimization problem over $\mathcal{E}^s_{v_i}$. This module leverages the GSyE at each node to learn its contextual information, thereby determining the expert assignments most suitable for each node. On a micro level, the GSyE optimizes subgraph Synergy by integrating structural details. On a global macro level, nodes with similar contextual characteristics across clients tend to preserve more similar and refined structures, promoting consistency from the node level. The specific process is as follows:

$$\boldsymbol{z} = \mathbf{X} \cdot \mathbf{W}_{\text{gate}}, \quad \mathbf{X} \in R^{N \times D}, \tag{12}$$

where $N$ is the number of nodes, and $D$ represents the dimensionality of the node features. $\boldsymbol{z}$ serves as the input for sampling from the hard concrete distribution (Louizos et al., 2017; Maddison et al., 2016; Schlichtkrull et al., 2020).

$$\psi(\boldsymbol{z}) = \alpha \cdot \sigma \left( \frac{\log(\boldsymbol{z}) - \log(1 - \boldsymbol{z}) + g_1 - g_2}{\tau} \right). \tag{13}$$

The term $\sigma$ denotes the Sigmoid function, $\alpha$ is a mixing coefficient, and $\tau$ is the temperature parameter controlling the smoothness of the distribution. $g_1$ and $g_2$ are random noise variables sampled for the reparameterization trick, which ensures differentiability of the binary gating mechanism.

$$\boldsymbol{z}' = \text{HardStep}(\psi(\boldsymbol{z})) = \begin{cases} 1, & \text{if } \psi(\boldsymbol{z}) > 0.5. \\ 0, & \text{otherwise.} \end{cases} \tag{14}$$

$\boldsymbol{z}'$ is a threshold function used to determine whether an expert is activated. Furthermore, we obtain the control matrix:

$$\mathbf{M}_{ij} = \begin{cases} \boldsymbol{z}'_{ij}, & \text{if } v_j \in \mathcal{G}^s_{v_i}. \\ 0, & \text{otherwise.} \end{cases} \tag{15}$$

Based on this, we define the calculation methods for expert importance scores and expert load:

$$\mathcal{I}^{imp.}_j = \sum_{i=1}^{N_{v_i}} \mathbf{M}_{ij}, \quad \mathcal{I}^{Loa.}_j = \frac{\mathcal{I}^{imp.}_j}{N_{v_i}}. \tag{16}$$

The optimization objective of the GSyE module and the procedure for optimizing $\mathbf{W}_{\text{gate}}$ are formulated as follows:

$$\mathcal{L}_{GSyE} = \lambda \cdot \left( \frac{Var(\mathcal{I}^{imp.})}{(\mathcal{I}^{imp.})^2 + \epsilon} \right) + (1 - \lambda) \cdot \left( \frac{Var(\mathcal{I}^{Loa.})}{(\mathcal{I}^{Loa.})^2 + \epsilon} \right),$$

$$\frac{\partial \mathcal{L}}{\partial \mathbf{W}_{\text{gate}}} = \frac{\partial \mathcal{L}}{\partial \boldsymbol{z}'} \cdot \frac{\partial \boldsymbol{z}'}{\partial \psi(\boldsymbol{z})} \cdot \frac{\partial \psi(\boldsymbol{z})}{\partial \boldsymbol{z}} \cdot \frac{\partial \boldsymbol{z}}{\partial \mathbf{W}_{\text{gate}}}, \tag{17}$$

where *Var* denotes the variance, $\lambda$ serves as a balancing coefficient to regulate the trade-off between the variance of importance scores and load scores, and $\epsilon$ is a small constant added to avoid division by zero. The primary node prediction task uses the standard negative log-likelihood loss:

$$\mathcal{L}_{NLL} = -\frac{1}{N} \sum_{i=1}^{N} \log(p(y_i|x_i)). \tag{18}$$

The final loss function comprises three components and is formulated as:

$$\mathcal{L}_{tot.} = \lambda_1 \cdot \mathcal{L}_{NLL} + \lambda_2 \cdot \mathcal{L}_{GSyE} + \lambda_3 \cdot \mathcal{L}_{Para}, \tag{19}$$

where $\lambda_i$ controls the relative weight of each loss component. $\mathcal{L}_{GSyE}$ and $\mathcal{L}_{Para}$ are defined in Equation (17) and Equation (6), respectively. $\lambda_3$ is applied only during pretraining for subnetwork training, and is set to $\lambda_3 = 0$ during retraining.

**Transport-based Similarity.** During each communication round, local clients upload their GNN parameters along with $\mathbf{W}_{\text{gate}}$ from the GSyE. The matrix $\mathbf{W}_{\text{gate}}$ encodes structural information derived from each client's data, as detailed in Section 4.3. The OT distance between clients $C_i$ and $C_j$ is then defined as:

$$d_{ij} = \min_{\mathbf{T} \in \mathcal{T}(a,b)} \sum_{m,n} \mathbf{T}_{mn} \cdot \mathbf{C}_{mn}, \tag{20}$$

where $a$ and $b$ denote the distributions derived from the unfolded $\mathbf{W}_{\text{gate}}$ matrices of clients $C_i$ and $C_j$, respectively. Here, $\mathcal{T}(a, b)$ is the set of feasible transport plans that satisfy supply and demand constraints. $\mathbf{T}_{mn}$ represents the amount of mass transported from the $m$-th element of distribution $a$ to the $n$-th element of distribution $b$, while $\mathbf{C}_{mn}$ denotes the unit transportation cost between these elements.

**Structure-Similarity Personalized Aggregation.** We leverage the OT distance between clients, formulated as:

$$d^{\text{norm}}_{ij} = \frac{d_{ij} - \min(\mathbf{d})}{\max(\mathbf{d}) - \min(\mathbf{d}) + \epsilon}, \tag{21}$$

where $\mathbf{d}$ represents the set of distance values between all pairs of clients. The similarity and personalized aggregation weight between clients $C_j$ and $C_i$ are defined as follows:

$$\mathbf{S}_{ij} = 1 - d^{norm}_{ij}, \quad W_{ij} = \frac{\mathbf{S}_{ij}}{\sum_{k=1}^{K} \mathbf{S}_{ik} + \epsilon}. \tag{22}$$

The parameter update rule for client $C_i$ is formulated as:

$$\theta^{t+1}_i = \sum_{j=1}^{K} W^t_{ij} \cdot \theta^t_j. \tag{23}$$

### 4.4. Discussion

Our method employs a unified sparsification framework to significantly reduce computational demands and communication overhead in federated systems. The parameter sparsification module identifies subnetworks across all sparsity levels in a single pass, albeit with a pretraining step inherited from conventional lottery

Table 1: Node classification performance comparison across multiple metrics with various methods from state-of-the-art federated systems and centralized environments. All reported results represent the mean over five runs. Green arrows ↑↓ indicate advancements in the given metric, while red arrows ↑↓ denote regressions. The best and second-best results are highlighted in bold and underlined, respectively. Please refer to section 5.2 for additional analysis. Additional experimental results on more datasets can be found in Appendix D.

| Methods | Effe. | Effi. | Econ. | Pubmed - (GCN) Top-1 Accuracy (↑) | Max Training FLOPS (↓) | Communication BYTES (↓) | Ogbn-Arxiv - (GraphSAGE) Top-1 Accuracy (↑) | Max Training FLOPS (↓) | Communication BYTES (↓) | Ogbn-Proteins - (DeeperGCN) ROC-AUC (↑) | Max Training FLOPS (↓) | Communication BYTES (↓) |
|---|---|---|---|---|---|---|---|---|---|---|---|---|
| FedAvg [ASTAT17] | ✗ | ✗ | ✗ | 85.65 | 1x(2.49E9) | 1x(6.19E9) | 63.99 | 1x(1.58E10) | 1x(6.14E9) | 81.54 | 1x(1.81E9) | 1x(2.26E9) |
| FedProx [arXiv18] | ✗ | ✗ | ✗ | $84.51_{\downarrow1.14}$ | $1.00x_{\uparrow0.00x}$ | $1.00x_{\downarrow0.00x}$ | $64.05_{\uparrow0.06}$ | $1.00x_{\uparrow0.00x}$ | $1.00x_{\downarrow0.00x}$ | $81.12_{\downarrow0.42}$ | $1.00x_{\uparrow0.00x}$ | $1.00x_{\downarrow0.00x}$ |
| FedSage+ [NeurIPS21] | ✓ | ✓ | ✗ | – | – | – | $64.23_{\uparrow0.24}$ | $1.00x_{\downarrow0.81x}$ | $1.00x_{\downarrow0.00x}$ | – | – | – |
| APPLE [IJCAI22] | ✗ | ✗ | ✗ | $83.23_{\downarrow2.32}$ | $1.00x_{\uparrow0.00x}$ | $1.00x_{\downarrow0.00x}$ | $63.21_{\downarrow0.78}$ | $1.00x_{\uparrow0.00x}$ | $1.00x_{\uparrow0.00x}$ | $80.02_{\downarrow1.52}$ | $1.00x_{\uparrow0.00x}$ | $1.00x_{\downarrow0.00x}$ |
| FedCP [KDD23] | ✓ | ✗ | ✗ | $85.82_{\uparrow0.17}$ | $1.03x_{\uparrow0.03x}$ | $1.17x_{\uparrow0.17x}$ | $64.11_{\uparrow0.12}$ | $3.22x_{\uparrow2.22x}$ | $1.22x_{\uparrow0.22x}$ | $81.49_{\downarrow0.05}$ | $3.86x_{\uparrow2.86x}$ | $1.28x_{\uparrow0.28x}$ |
| FedTAD [IJCAI24] | ✓ | ✗ | ✗ | $86.15_{\uparrow0.50}$ | $4.22x_{\uparrow3.22x}$ | $1.05x_{\uparrow0.05x}$ | $\underline{65.03}_{\uparrow1.04}$ | $35.33x_{\uparrow34.33x}$ | $1.13x_{\uparrow0.13x}$ | $\underline{82.02}_{\uparrow0.48}$ | $31.21x_{\uparrow30.21x}$ | $1.15x_{\uparrow0.15x}$ |
| FedSSP [NeurIPS24] | ✓ | ✗ | ✗ | $86.32_{\uparrow0.67}$ | $3.21x_{\uparrow2.21x}$ | $2.45x_{\uparrow1.45x}$ | $64.34_{\uparrow0.35}$ | $27.89x_{\uparrow26.89x}$ | $2.33x_{\uparrow1.33x}$ | $81.98_{\uparrow0.44}$ | $25.78x_{\uparrow24.78x}$ | $2.89x_{\uparrow1.89x}$ |
| FGGP [AAAI24] | ✗ | ✗ | ✗ | $84.55_{\uparrow1.10}$ | $1.15x_{\uparrow0.15x}$ | $1.32x_{\uparrow0.32x}$ | $63.78_{\downarrow0.21}$ | $5.78x_{\uparrow4.78x}$ | $1.44x_{\uparrow0.44x}$ | $80.56_{\downarrow0.98}$ | $11.23x_{\uparrow10.23x}$ | $7.65x_{\uparrow6.65x}$ |
| FedGTA [VLDB 24] | ✓ | ✗ | ✗ | $\underline{86.45}_{\uparrow0.80}$ | $1.00x_{\uparrow0.00x}$ | $1.00x_{\uparrow0.00x}$ | $64.13_{\uparrow0.14}$ | $1.00x_{\uparrow0.00x}$ | $1.00x_{\uparrow0.00x}$ | $81.78_{\uparrow0.24}$ | $1.00x_{\uparrow0.00x}$ | $1.00x_{\uparrow0.00x}$ |
| PruneFL [TNNLS19] | ✗ | ✓ | ✗ | $80.45_{\downarrow5.11}$ | $0.67x_{\downarrow0.33x}$ | $1.00x_{\downarrow0.00x}$ | $60.88_{\downarrow3.11}$ | $0.55x_{\downarrow0.45x}$ | $1.00x_{\downarrow0.00x}$ | $78.89_{\downarrow2.65}$ | $0.66x_{\downarrow0.34x}$ | $1.00x_{\downarrow0.00x}$ |
| FedTiny [ICDCS23] | ✗ | ✓ | ✓ | $82.84_{\downarrow2.81}$ | $0.77x_{\downarrow0.23x}$ | $0.80x_{\downarrow0.20x}$ | $61.98_{\downarrow2.01}$ | $0.51x_{\downarrow0.49x}$ | $0.69x_{\downarrow0.31x}$ | $79.92_{\downarrow1.62}$ | $\underline{0.55x}_{\downarrow0.45x}$ | $0.59x_{\downarrow0.41x}$ |
| FedDIP [ICDM23] | ✗ | ✓ | ✓ | $83.30_{\downarrow2.35}$ | $0.59x_{\downarrow0.41x}$ | $\underline{0.56x}_{\downarrow0.44x}$ | $63.85_{\downarrow0.14}$ | $0.38x_{\downarrow0.62x}$ | $\underline{0.58x}_{\downarrow0.42x}$ | $81.21_{\downarrow0.33}$ | $0.56x_{\downarrow0.44x}$ | $\underline{0.42x}_{\downarrow0.58x}$ |
| DSpar [TMLR23] | ✗ | ✗ | ✗ | $84.14_{\downarrow1.51}$ | $1.00x_{\downarrow0.00x}$ | $1.00x_{\downarrow0.00x}$ | $63.44_{\downarrow0.55}$ | $1.00x_{\downarrow0.00x}$ | $1.00x_{\downarrow0.00x}$ | $80.86_{\downarrow0.68}$ | $1.00x_{\downarrow0.00x}$ | $1.00x_{\downarrow0.00x}$ |
| ACE-GLT [CVPR24] | ✓ | ✓ | ✗ | $85.68_{\uparrow0.03}$ | $\underline{0.55x}_{\downarrow0.45x}$ | $1.00x_{\downarrow0.00x}$ | $64.04_{\uparrow0.05}$ | $\underline{0.36x}_{\downarrow0.64x}$ | $1.00x_{\downarrow0.00x}$ | $81.58_{\uparrow0.04}$ | $0.59x_{\downarrow0.41x}$ | $1.00x_{\downarrow0.00x}$ |
| EAGLES | ✓ | ✓ | ✓ | $\mathbf{86.97}_{\uparrow1.32}$ | $\mathbf{0.48x}_{\downarrow0.52x}$ | $\mathbf{0.37x}_{\downarrow0.63x}$ | $\mathbf{65.37}_{\uparrow1.38}$ | $\mathbf{0.32x}_{\downarrow0.68x}$ | $\mathbf{0.48x}_{\downarrow0.52x}$ | $\mathbf{82.10}_{\uparrow0.56}$ | $\mathbf{0.18x}_{\downarrow0.82x}$ | $\mathbf{0.20x}_{\downarrow0.80x}$ |

networks. This pretraining, however, enables substantial parameter sparsification, greatly reducing communication costs during federated updates. At the graph sparsification level, a multi-expert approach is introduced to effectively mitigate *structure information overfitting*, a common issue in single-standard sparsification methods.

## 5. Experiments

In this section, we omprehensively evaluate our proposed EAGLES by addressing the following key questions.

- **Q1: Superiority.** Does EAGLES maintain or surpass baseline performance?
- **Q2: Efficiency and Economical.** Is EAGLES capable of reducing computational resource requirements and communication bytes?
- **Q3: Resilience.** What is the performance of EAGLES under varying sparsification rates and different numbers of clients?
- **Q4: Sensitivity.** How does EAGLES perform with different hyper-parameter settings?

The answer of **Q1-Q3** are illustrated in 5.2-5.4, and the analyses of **Q4** can be found in Appendix D.

### 5.1. Experiment Setup

**Datasets and Split.** To comprehensively evaluate EA-GLES across different datasets and tasks, we selected Cora (Mammen, 2021), Pubmed (Shchur et al., 2018), and Photo (McAuley et al., 2015) from small to medium-scale datasets, and Ogbn-Arxiv, Ogbn-Proteins, and Ogbn-Products from large-scale datasets (Hu et al., 2020). Detailed descriptions for these datasets can be found in Appendix A.

**Backbone and Parameter Configurations.** For small to medium-scale datasets, we use a two-layer GCN as the back-

bone, while for large-scale datasets, we select a four-layer GraphSAGE and DeeperGCN as the backbones (Defferrard et al., 2016; Li et al., 2018; Zhang et al., 2018). Please see Appendix E for more details.

**Evaluation Details.** We utilize a range of metrics to comprehensively evaluate the multi-faceted performance of each method: ❶ **T**op-1 **Ac**curacy (Top-1-AC). ❷ **R**eceiver **O**perating **C**haracteristic-**A**rea **U**nder the **C**urve (ROC-AUC) (Bradley, 1997). ❸ **M**ax **T**raining **F**LOPS (TF). ❹ **C**ommunication **B**ytes (CB). We compared our method with traditional FL methods: (1) FedAvg (McMahan et al., 2017); (2) FedProx (Li et al., 2020); (3) FedSage+ (Zhang et al., 2021); six state-of-the-art FGL methods: (4) APPLE (Luo & Wu, 2022); (5) FedCP (Zhang et al., 2023); (6) FedTAD (Zhu et al., 2024); (7) FedSSP (Tan et al., 2024); (8) FGGP (Wan et al., 2024a); (9) FedTGA (Li et al., 2024); two widely used graph sparsification methods: (10) DSpar (Liu et al., 2023c); (11) ACE-GLT (Wang et al., 2023); and three recent pruning methods for traditional FL: (12) FedDIP (Long et al., 2023); (13) PruneFL (Jiang et al., 2022); (14) FedTiny (Huang et al., 2023). Additional evaluation details are provided in Appendix B.

### 5.2. Superiority

To address **Q1**, we compare EAGLES against state-of-the-art methods in federated and centralized settings across multiple metrics, as shown in Table 1. Most existing methods excel in specific metrics but fail to achieve overall superiority: ❶ FedCP, FedTAD, and FedSSP improve model generalization but neglect the redundancy in graph data and model parameters, which hampers training quality and limits potential gains. ❷ Graph sparsification and model pruning methods, such as DSpar, reduce computational

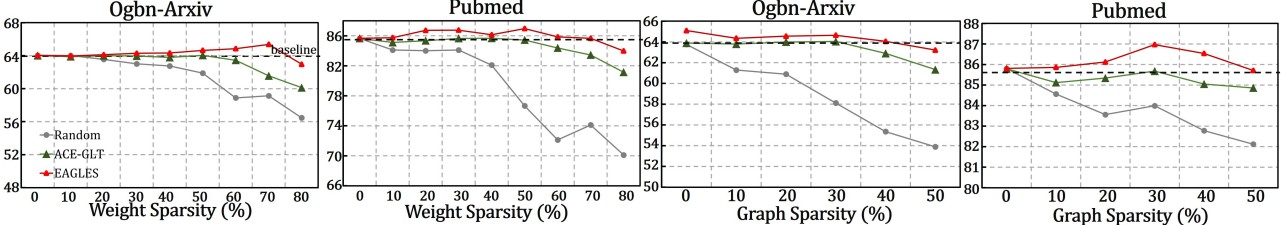

Figure 4: Node classification performance comparison of accuracy on the Pubmed and Ogbn-arxiv datasets, illustrating results for random pruning, EAGLES, and the advanced dual-sparsification approach ACE-GLT. Please see details in section 5.4.

costs but often compromise model performance due to simplistic criteria like single-factor sparsification, leading to structural overfitting. ❸ FedTiny adopts non-recoverable pruning without considering parameter importance across clients, resulting in significant performance degradation. In contrast, EAGLES achieves superior performance across all metrics, effectively minimizing Training FLOPS and Communication BYTES while maintaining or even enhancing model performance.

### 5.3. Efficiency

To address **Q2**, we examine Training FLOPS and Communication BYTES, as summarized in Table 1, with metric details provided in Appendix B. Methods like FedSSP and FedTAD improve local or global generalization but at the cost of higher computational and communication overhead, particularly in large-scale graphs. Sparsification approaches, such as PruneFL and ACE-GLT, involve weight recovery, preventing reductions in communication bytes. Moreover, PruneFL lacks graph-level sparsification, and both methods overlook client-specific parameter importance, limiting their contributions to FGL. FedTiny and FedDIP avoid weight recovery by pruning during training, reducing both Training FLOPS and Communication BYTES, but at the expense of model performance. In contrast, EAGLES employs consensus-based parameter sparsification to identify subnetworks at each pruning rate, leveraging these subnetworks during retraining to simultaneously reduce dataset size and ensure both efficiency and effectiveness. Furthermore, the reduced parameter size of the subnetworks significantly lowers communication bytes, achieving a more economical training process.

### 5.4. Resilience

To address **Q3**, we conducted extensive experiments. Table 2 and Table 4 present the Top-1 Accuracy of EAGLES across various sparsification rates. **Observations: ❶** The optimal graph sparsification rate is approximately 30%, while parameter sparsification shows greater flexibility. For example, under the Ogbn-Proteins + DeeperGCN configuration, a parameter sparsification rate of 81.79% (rounded to 80% in the table) results in a 0.4% performance improvement over the baseline. We present ablation experiments on the number of clients in Figure 6a and Figure 6b.

Figure 4 compares EAGLES with ACE-GLT, a state-

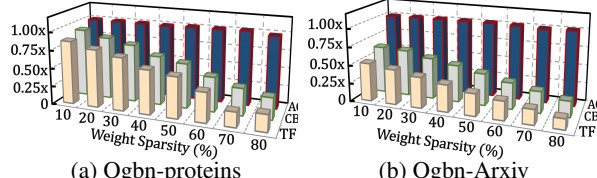

(a) Ogbn-proteins    (b) Ogbn-Arxiv

Figure 5: **Three-dimensional bar chart** delineates the influence of weight sparsity on three key metrics: AC (Top-1 Accuracy), CB (Communication BYTES), and TF (Training FLOPS). For an in-depth analysis, please refer to section 5.4.

of-the-art dual-sparsification method for centralized systems, and random sparsification on node classification tasks for Pubmed and Ogbn-Arxiv. For weight sparsity analysis, graph sparsity is fixed at 30%; for graph sparsity analysis, weight sparsity is fixed at 50%. **Observations: ❷** EAGLES consistently outperforms the baseline under most conditions, while ACE-GLT falls short. This discrepancy likely arises from ACE-GLT's limited consideration of data heterogeneity in federated systems and the varying importance of parameters across clients.

Figure 5 presents two **three-dimensional bar charts** illustrating three key metrics—AC (Top-1 Accuracy), CB (Communication BYTES), and TF (Training FLOPS)—on the Ogbn-Arxiv and Ogbn-Proteins datasets. Baseline metrics are normalized to 1.00x, with all results reported as percentages relative to this baseline. We observe that **Obs. ❸** as weight sparsity increases, both TF and CB steadily decrease, while AC remains stable, consistently near or above the baseline. Collectively, these observations provide strong evidence of the resilience of EAGLES.

### 6. Conclusion

In this paper, we are pioneers in addressing the challenge of reducing computational resource demands in federated graph learning. We introduce **EAGLES**: Towards **E**ffective, Efficient, **A**nd Economical Federated **G**raph **L**earning via Unifi**E**d **S**parsification. Previous works rely on sparsification in a single aspect, we are the first to propose unified sparsification, overcoming issues such as parameter space sparsification without federated characteristics and graph sparsification constrained by single criteria. Our method's superiority is demonstrated across multiple metrics on several datasets. We hope this work offers a novel perspective for future research focused on reducing computational resource consumption in federated graph systems.

## Acknowledgement

This research is supported by the National Key Research and Development Project of China (2024YFC3308400), the National Natural Science Foundation of China (Grants 62361166629, 62176188, 623B2080), the Wuhan University Undergraduate Innovation Research Fund Project. The supercomputing system at the Supercomputing Center of Wuhan University supported the numerical calculations in this paper. Carl Yang was not supported by any funds from China.

## Impact Statement

This paper presents work whose goal is to advance the field of Machine Learning. There are many potential societal consequences of our work, none of which we feel must be specifically highlighted here.

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

# A. Datasets Details

All datasets are used for node classification tasks. For the small to medium group Cora, pubmed and Amz-Photo, we split training, validation and test set as 60%, 20%, 20%. we manually split the data into 60% for training, 20% for validation, and 20% for testing, while for the latter group, we utilized the official dataset splits. The official (training, validation, test) splits for OGBN-Arxiv, OGBN-Proteins, and OGBN-Products are (53.7%, 17.6%, 28.7%), (65.3%, 13.9%, 20.8%), and (8.03%, 4.01%, 87.96%), respectively. The statistics of the datasets used in our experiments are provided in Table 5.

- **Citation Network** (Cora and pubmed): The Citation Network datasets Cora and pubmed are standard benchmarks for graph neural networks in machine learning. The Cora dataset contains 2,708 scientific publications divided into seven categories. Each publication is linked by citations and represented by a binary word vector from a vocabulary of 1,433 unique words. The pubmed dataset focuses on diabetes and includes 19,717 publications across three categories: Diabetes Mellitus, Experimental Diabetes Mellitus Type 1, and Diabetes Mellitus Type 2. It has 44,338 citation links, with each publication represented by a TF-IDF weighted word vector from a dictionary of 500 unique terms (Yang et al., 2016).

- **Amz-purchase** (Photo): The Amz-purchase Photo dataset, derived from Amazon's co-purchase network, is widely utilized for classification tasks in recommendation systems and market analysis. In this dataset, nodes represent products, while edges indicate co-purchase relationships, which capture products often bought together. Each product is associated with specific features, and the main task is to predict its category. This dataset supports research on consumer behavior, product categorization, and recommendation algorithms (McAuley et al., 2015).

- **OGBN-ARXIV** The OGBN-ARXIV dataset from the Open Graph Benchmark (OGB) supports node property prediction tasks within an academic citation network. This dataset comprises 169,343 nodes, each representing a computer science paper from the arXiv repository with directed edges indicating citation relationships. Each node includes a 128-dimensional feature vector derived from embeddings of the paper's title and abstract. The main task is to predict the paper's subject area across 40 classes. The dataset is partitioned by publication year: papers published up to 2017 are used for training, those from 2018 for validation, and those from 2019 onward for testing (Hu et al., 2020).

- **OGBN-PROTEINS** The OGBN-PROTEINS dataset from the Open Graph Benchmark (OGB) supports node property prediction tasks within a protein-protein association network. This undirected weighted graph contains 132,534 nodes, each representing a protein from one of eight species. Edges indicate biologically meaningful associations such as physical interactions, co-expression, or homology. Each edge has an 8-dimensional feature vector where each dimension represents the confidence score (from 0 to 1) of a specific association type. The primary task is to predict 112 different protein functions in a multi-label binary classification setup. The dataset is split by species, allowing models to be evaluated on their generalization across species (Hu et al., 2020).

- **OGBN-PRODUCTS** The OGBN-PRODUCTS dataset from the Open Graph Benchmark (OGB) supports node property prediction tasks within an Amazon product co-purchasing network. This undirected unweighted graph includes approximately 2.4 million nodes, each representing a product, and 61.9 million edges indicating frequent co-purchase relationships. Each node has a 100-dimensional feature vector derived from the product description. The primary task is to predict the product category among 47 top-level classes in a multi-class classification setup. The dataset is split by sales rank, with the top 8% of products for training, the next 2% for validation, and the remaining 90% for testing, simulating a realistic scenario where popular products have labeled data while predictions are required for less popular items (Hu et al., 2020).

# B. More Evaluation Details

The metrics used in the experiments in this paper are calculated as follows:

- **Top-1 Accuracy** Top-1 Accuracy is a common evaluation metric used in classification tasks, particularly for multi-class problems. It represents the proportion of predictions where the model's highest confidence label (the top-1 prediction) matches the true label. This metric is calculated as the ratio of correctly classified samples to the total number of samples, providing a straightforward measure of the model's ability to correctly identify the primary class for each input.

$$\textbf{Accuracy} = \frac{1}{T} \sum_{i=1}^{T} \frac{\sum_{j=1}^{N} \mathbf{1}\{y_{\text{true}}[j, i] = y_{\text{pred}}[j, i]\}}{N} \tag{24}$$

Table 2: Node classification performance of EAGLES evaluated across numerous values of $S^{\mathcal{G}}$ and $S^{\mathbf{W}}$ on pubmed, Ogbn-Arxiv and Ogbn-Proteins. The reported performance represents the average over five runs and is measured within a target sparsity range of ±3%. Please refer to appendix D for additional analysis.

| Parameter Sparsity% | Graph Sparsity% - pubmed | | | | Graph Sparsity% - Ogbn-Arxiv | | | | Graph Sparsity% - Ogbn-Proteins | | | |
|---|---|---|---|---|---|---|---|---|---|---|---|---|
| | 0 | 10 | 30 | 50 | 0 | 10 | 30 | 50 | 0 | 10 | 30 | 50 |
| 0 | $84.88_{\downarrow 0.77}$ | $85.50_{\downarrow 0.15}$ | $85.71_{\uparrow 0.06}$ | $84.98_{\downarrow 0.67}$ | $63.45_{\downarrow 0.54}$ | $63.32_{\downarrow 0.67}$ | $64.01_{\downarrow 0.02}$ | $62.89_{\downarrow 1.10}$ | $81.89_{\uparrow 0.35}$ | $82.13_{\uparrow 0.59}$ | $81.92_{\uparrow 0.38}$ | $81.88_{\uparrow 0.34}$ |
| 10 | $84.65_{\downarrow 1.00}$ | $85.51_{\downarrow 0.14}$ | $85.75_{\uparrow 0.10}$ | $84.88_{\downarrow 0.77}$ | $63.23_{\downarrow 0.76}$ | $63.48_{\downarrow 0.51}$ | $63.97_{\uparrow 0.02}$ | $62.57_{\downarrow 1.42}$ | $81.84_{\uparrow 0.30}$ | $82.11_{\uparrow 0.57}$ | $81.90_{\uparrow 0.36}$ | $81.97_{\uparrow 0.43}$ |
| 20 | $84.66_{\downarrow 0.99}$ | $85.79_{\uparrow 0.14}$ | $86.73_{\uparrow 1.08}$ | $84.27_{\downarrow 1.38}$ | $63.11_{\downarrow 0.88}$ | $63.88_{\downarrow 0.11}$ | $64.11_{\uparrow 0.12}$ | $62.66_{\downarrow 1.33}$ | $82.97_{\uparrow 1.43}$ | $82.62_{\uparrow 1.08}$ | $82.22_{\uparrow 0.68}$ | $81.70_{\uparrow 0.16}$ |
| 30 | $85.66_{\uparrow 0.01}$ | $86.99_{\uparrow 1.34}$ | $86.76_{\uparrow 1.11}$ | $84.24_{\downarrow 1.41}$ | $64.42_{\uparrow 0.43}$ | $64.12_{\uparrow 0.13}$ | $64.05_{\uparrow 0.06}$ | $63.92_{\downarrow 0.07}$ | $82.82_{\uparrow 1.28}$ | $83.76_{\uparrow 2.22}$ | $82.69_{\uparrow 1.15}$ | $82.61_{\uparrow 1.07}$ |
| 40 | $85.12_{\downarrow 0.53}$ | $84.55_{\downarrow 1.10}$ | $86.14_{\uparrow 0.49}$ | $84.38_{\downarrow 1.27}$ | $64.95_{\uparrow 0.96}$ | $64.59_{\uparrow 0.60}$ | $64.35_{\uparrow 0.36}$ | $63.66_{\downarrow 0.33}$ | $82.62_{\uparrow 1.08}$ | $82.29_{\uparrow 0.75}$ | $83.05_{\uparrow 1.51}$ | $82.54_{\uparrow 1.00}$ |
| 50 | $85.81_{\uparrow 0.16}$ | $85.88_{\uparrow 0.23}$ | $86.97_{\uparrow 1.32}$ | $85.71_{\uparrow 0.06}$ | $65.09_{\uparrow 1.10}$ | $64.35_{\uparrow 0.36}$ | $64.64_{\uparrow 0.65}$ | $63.21_{\downarrow 0.78}$ | $81.47_{\downarrow 0.07}$ | $82.18_{\uparrow 0.64}$ | $83.24_{\uparrow 1.70}$ | $81.44_{\downarrow 0.10}$ |
| 60 | $84.89_{\downarrow 0.76}$ | $85.44_{\downarrow 0.21}$ | $85.86_{\uparrow 0.21}$ | $82.78_{\downarrow 2.87}$ | $62.63_{\downarrow 1.36}$ | $63.02_{\downarrow 0.97}$ | $64.85_{\uparrow 0.86}$ | $64.08_{\uparrow 0.09}$ | $82.63_{\uparrow 1.09}$ | $82.25_{\uparrow 0.71}$ | $82.64_{\uparrow 1.10}$ | $82.52_{\uparrow 0.98}$ |
| 70 | $83.88_{\downarrow 1.77}$ | $85.78_{\downarrow 0.13}$ | $85.67_{\uparrow 0.02}$ | $83.19_{\downarrow 2.46}$ | $64.19_{\uparrow 0.20}$ | $64.20_{\uparrow 0.21}$ | $65.37_{\uparrow 1.38}$ | $63.34_{\downarrow 0.65}$ | $81.58_{\uparrow 0.04}$ | $81.29_{\downarrow 0.25}$ | $81.55_{\uparrow 0.01}$ | $81.27_{\downarrow 0.27}$ |
| 80 | $82.28_{\downarrow 3.37}$ | $83.89_{\downarrow 1.76}$ | $83.98_{\downarrow 1.67}$ | $82.17_{\downarrow 3.48}$ | $61.28_{\downarrow 2.71}$ | $60.41_{\downarrow 3.58}$ | $62.96_{\downarrow 1.03}$ | $60.13_{\downarrow 3.86}$ | $81.94_{\uparrow 0.40}$ | $80.50_{\downarrow 1.04}$ | $82.10_{\uparrow 0.56}$ | $80.87_{\downarrow 0.67}$ |
| Vanilla | 85.65 | | | | 63.99 | | | | 81.54 | | | |

Table 3: **Statistics** of datasets used in experiments.

| Dataset | #Nodes | #Edges | #Classes | #Average Degree | #Metric |
|---|---|---|---|---|---|
| Cora | 2,708 | 5,278 | 7 | 2.25 | Accuracy |
| pubmed | 19,717 | 44,324 | 3 | 8.00 | Accuracy |
| Amz-Photo | 7,650 | 287,326 | 8 | 31.12 | Accuracy |
| Ogbn-Arixv | 169,343 | 1,166,243 | 40 | 13.77 | Accuracy |
| Ogbn-Proteins | 132,534 | 39,561,252 | 2 | 597.00 | ROC-AUC |
| Ogbn-Products | 2,449,029 | 61,859,140 | 47 | 50.52 | Accuracy |

- **ROC-AUC** The ROC-AUC is a widely adopted metric for evaluating binary classifiers and can be extended to multi-label settings. The ROC curve plots the true positive rate (TPR) against the false positive rate (FPR) across various thresholds, illustrating the model's performance over a range of decision boundaries. The AUC (Area Under the Curve) quantifies the likelihood that a randomly chosen positive instance ranks above a randomly chosen negative instance. An AUC of 1.0 indicates perfect discrimination, while an AUC of 0.5 suggests performance no better than random guessing.

$$\text{ROC-AUC} = \frac{1}{T} \sum_{i=1}^{T} \int_0^1 \text{TPR}_i \left( \text{FPR}_i^{-1}(t) \right) \, dt,$$
$$\text{TPR} = \frac{\text{TP}}{\text{TP} + \text{FN}}, \quad \text{FPR} = \frac{\text{FP}}{\text{FP} + \text{TN}}.$$
(25)

where TP (True Positives) denotes correctly classified positive samples, FN (False Negatives) represents positive samples misclassified as negative, FP (False Positives) indicates negative samples misclassified as positive, and TN (True Negatives) is the count of correctly classified negative samples. These values are essential for calculating the True Positive Rate (TPR) and False Positive Rate (FPR), which are used to plot the ROC curve and compute the Area Under the Curve (ROC-AUC).

- **MAX Training FLOPS** The calculation of MAX Training FLOPs primarily addresses two main components: message-passing operations and feature transformation. The message-passing aspect accounts for the computational load of propagating information across nodes and edges, while the feature transformation component captures operations within masked linear layers and activation functions such as ReLU and BatchNorm (with Mixture of Experts (MoE) modules in the case of EAGLES). This approach provides a comprehensive assessment of the FLOPs required for both propagation and transformation processes within the model.

- **Communication BYTES** Communication cost is calculated based on the total bytes transmitted by clients to the server (upload bytes) and the bytes required for the server to broadcast the aggregated global model back to each client (download bytes). The model size serves as the basis for calculating both upload and download bytes, with each client transmitting its model to the server and receiving the updated global model after aggregation. This metric reflects the overall data transfer required to synchronize model updates across the network.

Table 4: Node classification performance of EAGLES evaluated across numerous values of $S^{\mathcal{G}}$ and $S^{\mathbf{W}}$ on Cora, photo and Ogbn-Products. Please refer to appendix D for additional analysis.

| Parameter Sparsity% | Graph Sparsity% - Cora | | | | Graph Sparsity% - Photo | | | | Graph Sparsity% - Ogbn-Products | | | |
|---|---|---|---|---|---|---|---|---|---|---|---|---|
| | 0 | 10 | 30 | 50 | 0 | 10 | 30 | 50 | 0 | 10 | 30 | 50 |
| 0 | $74.89_{\downarrow 0.06}$ | $75.11_{\uparrow 0.16}$ | $74.77_{\downarrow 0.18}$ | $73.55_{\downarrow 1.40}$ | $92.03_{\uparrow 0.01}$ | $85.51_{\downarrow 0.14}$ | $85.75_{\uparrow 0.10}$ | $84.88_{\downarrow 0.77}$ | $73.82_{\downarrow 0.04}$ | $73.81_{\downarrow 0.05}$ | $73.66_{\downarrow 0.20}$ | $72.75_{\downarrow 1.11}$ |
| 10 | $76.08_{\uparrow 1.13}$ | $75.48_{\uparrow 0.53}$ | $75.11_{\uparrow 0.16}$ | $73.60_{\downarrow 1.35}$ | $92.86_{\uparrow 0.84}$ | $92.68_{\uparrow 0.66}$ | $92.52_{\uparrow 0.50}$ | $92.18_{\uparrow 0.16}$ | $73.65_{\downarrow 0.21}$ | $73.89_{\uparrow 0.03}$ | $73.99_{\uparrow 0.13}$ | $73.82_{\downarrow 0.04}$ |
| 20 | $76.23_{\uparrow 1.28}$ | $75.59_{\uparrow 0.64}$ | $74.77_{\downarrow 0.18}$ | $73.46_{\downarrow 1.49}$ | $92.40_{\uparrow 0.38}$ | $92.21_{\uparrow 0.19}$ | $92.54_{\uparrow 0.52}$ | $92.28_{\uparrow 0.26}$ | $73.88_{\downarrow 0.02}$ | $74.11_{\uparrow 0.25}$ | $74.13_{\uparrow 0.27}$ | $74.02_{\uparrow 0.16}$ |
| 30 | $75.55_{\uparrow 0.60}$ | $74.97_{\uparrow 0.02}$ | $74.60_{\downarrow 0.35}$ | $72.27_{\downarrow 2.68}$ | $92.04_{\uparrow 0.02}$ | $92.37_{\uparrow 0.35}$ | $92.42_{\uparrow 0.40}$ | $92.25_{\uparrow 0.23}$ | $73.81_{\downarrow 0.05}$ | $73.99_{\uparrow 0.13}$ | $74.22_{\uparrow 0.36}$ | $74.15_{\uparrow 0.29}$ |
| 40 | $76.14_{\uparrow 1.19}$ | $75.15_{\uparrow 0.20}$ | $75.10_{\uparrow 0.15}$ | $72.58_{\downarrow 2.37}$ | $92.46_{\uparrow 0.44}$ | $92.41_{\uparrow 0.39}$ | $92.31_{\uparrow 0.29}$ | $92.26_{\uparrow 0.24}$ | $74.23_{\uparrow 0.37}$ | $74.85_{\uparrow 0.99}$ | $74.98_{\uparrow 1.12}$ | $74.16_{\downarrow 0.30}$ |
| 50 | $75.30_{\uparrow 0.35}$ | $75.25_{\uparrow 0.30}$ | $74.43_{\downarrow 0.52}$ | $73.09_{\downarrow 1.86}$ | $92.26_{\uparrow 0.24}$ | $92.11_{\uparrow 0.09}$ | $91.95_{\downarrow 0.07}$ | $91.86_{\downarrow 0.16}$ | $74.33_{\uparrow 0.47}$ | $74.78_{\uparrow 0.92}$ | $74.46_{\uparrow 0.60}$ | $74.09_{\uparrow 0.23}$ |
| 60 | $74.85_{\downarrow 0.10}$ | $75.26_{\uparrow 0.31}$ | $74.31_{\downarrow 0.64}$ | $72.75_{\downarrow 2.20}$ | $92.40_{\uparrow 0.38}$ | $92.08_{\uparrow 0.06}$ | $92.49_{\uparrow 0.47}$ | $92.14_{\uparrow 0.12}$ | $73.99_{\uparrow 0.13}$ | $74.07_{\uparrow 0.21}$ | $73.76_{\downarrow 0.10}$ | $73.94_{\uparrow 0.08}$ |
| 70 | $75.23_{\uparrow 0.28}$ | $75.04_{\uparrow 0.09}$ | $74.88_{\downarrow 0.07}$ | $72.69_{\downarrow 2.26}$ | $92.23_{\uparrow 0.21}$ | $92.09_{\uparrow 0.07}$ | $92.75_{\uparrow 0.73}$ | $92.77_{\downarrow 0.75}$ | $73.88_{\downarrow 0.02}$ | $73.64_{\downarrow 0.22}$ | $73.93_{\uparrow 0.07}$ | $73.82_{\downarrow 0.04}$ |
| 80 | $73.94_{\downarrow 1.01}$ | $75.33_{\uparrow 0.38}$ | $72.46_{\downarrow 2.59}$ | $71.44_{\downarrow 3.51}$ | $91.88_{\downarrow 0.14}$ | $92.27_{\uparrow 0.25}$ | $92.23_{\uparrow 0.21}$ | $91.52_{\downarrow 0.50}$ | $73.52_{\downarrow 0.34}$ | $73.84_{\downarrow 0.02}$ | $72.65_{\downarrow 1.21}$ | $72.99_{\downarrow 0.87}$ |
| Vanilla | 74.95 | | | | 92.02 | | | | 73.86 | | | |

Table 5: Detailed hyper-parameter configurations.

| Dataset | #Model | #Round | #Weight Decay | #learning rate | #Optimizer |
|---|---|---|---|---|---|
| Cora | GCN | 200 | 2e-4 | 0.01 | Adam |
| pubmed | GCN | 500 | 2e-4 | 0.01 | Adam |
| Amz-Photo | GCN | 800 | 2e-4 | 0.01 | Adam |
| Ogbn-Arixv | GraphSAGE | 800 | 1e-6 | 0.01 | Adam |
| Ogbn-Proteins | DeeperGCN | 300 | 5e-6 | 0.01 | Adam |
| Ogbn-Products | GraphSAGE | 800 | 1e-6 | 0.01 | Adam |

In Table 1, the reported FLOPS and BYTES values for each sparsification baseline method correspond to the scenarios where the highest accuracy is achieved. And the details of the baseline employed for comparison are as follows:

- **FedAvg** (McMahan et al., 2017): FedAvg is a foundational algorithm in federated learning, allowing multiple clients to collaboratively train a global model without sharing their raw data. Each client locally computes model updates based on its data, and the server aggregates these updates through periodic averaging to form the global model. This process maintains data privacy and minimizes communication costs, making it efficient for distributed environments.

- **FedProx** (Li et al., 2020): FedProx extends FedAvg by introducing a proximal term in the objective function, which helps address issues related to client heterogeneity. By regularizing local updates toward the global model, FedProx stabilizes convergence, particularly in federated learning environments with non-iid data distributions and clients with diverse computational capabilities.

- **APPLE** (Luo & Wu, 2022): The Adaptive Personalized Cross-Silo Federated Learning (APPLE) framework addresses non-IID data challenges by enabling clients to personalize models through selective integration of others' model information. APPLE employs three components: Core Model Sharing, where clients share a *core model* with the server for distribution; Directed Relationship (DR) Vector, assigning weights to received core models to enhance relevant integration; and a Dynamic Penalty Mechanism that balances global and local objectives, allowing clients to tailor the model to both global trends and local needs.

- **FedSage+** (Zhang et al., 2021): FedSage+ is a federated learning framework designed for graph neural networks (GNNs), addressing challenges of distributed graph data and heterogeneity. The framework leverages *Dynamic Neighbor Sampling* to efficiently sample and aggregate information from local graph structures, reducing computational overhead while maintaining performance. FedSage+ employs three key components: Local Subgraph Training, where clients independently train on sampled subgraphs to preserve data privacy; Federated Aggregation, combining client models via weighted averaging (e.g., FedAvg); and Model Sparsification, reducing communication costs by pruning less important model parameters. This design achieves efficient learning while balancing global knowledge and local data diversity.

- **FedCP** (Zhang et al., 2023): The Federated Conditional Policy (FedCP) method addresses client data heterogeneity in federated learning by separating global and personalized information within feature representations. It employs a Conditional Policy Network (CPN) to generate sample-specific policies that partition features into global and personalized components. These components are then processed by a shared global head and a client-specific personalized head, respectively. This approach enables more fine-grained personalization, enhancing model performance across diverse client data distributions.

- **FedTAD** (Zhu et al., 2024): FedTAD is a topology-aware data-free knowledge distillation method designed for subgraph federated learning. It addresses subgraph heterogeneity by decoupling node and topology variations, which correspond to differences in label distribution and structure homophily. FedTAD enhances reliable knowledge transfer from local models to the global model by leveraging topology-aware node embeddings to measure class-wise knowledge reliability. This approach improves the performance of the global graph neural network without requiring raw data exchange, making it suitable for scenarios with data privacy constraints.

- **FedSSP** (Tan et al., 2024): FedSSP is a federated graph learning framework that addresses structural heterogeneity across clients by sharing generic spectral knowledge and accommodating personalized preferences. It introduces a global spectral knowledge-sharing mechanism to capture common structural patterns and a personalized preference module to adjust local message-passing schemes. This combination enhances model performance in cross-domain scenarios by effectively balancing global collaboration and local adaptation.

- **FGGP** (Wan et al., 2024a): Federated Graph Learning with Generalizable Prototypes (FGGP) addresses the challenge of domain shift in federated graph learning. It decouples the global model into two levels—feature extractor and classifier—and bridges them via prototypes, which serve as semantic centers derived from the feature extractor. At the classifier level, FGGP leverages clustered prototypes to capture domain-specific information and enhance class discriminability. At the feature extractor level, it employs contrastive learning to obtain more robust prototypes, thereby improving the generalization ability of the feature extractor across diverse domains.

- **FedGTA** (Li et al., 2024): FedGTA (Federated Graph Topology Augmentation) is a federated learning framework tailored for graph data, tackling challenges of graph heterogeneity and sparsity. The framework introduces *Topology Augmentation*, dynamically enhancing local graph structures by generating virtual neighbors or predicting missing edges based on node feature similarity and local connectivity. FedGTA employs three key components: Local Graph Enhancement, where clients augment their subgraphs to address sparsity; Federated Aggregation, combining model updates and augmentation summaries to improve global learning; and Personalized Topology Rules, allowing clients to adapt global trends to their unique graph structures. This approach enhances performance in distributed, heterogeneous graph scenarios while ensuring data privacy.

- **DSpar** (Liu et al., 2023c): DSpar offers a straightforward strategy for improving the efficiency of Graph Neural Network (GNN) training and inference by selectively sparsifying graph structures based on node degree. By prioritizing higher-degree nodes, DSpar reduces the number of edges while preserving essential structural information, thus significantly lowering computational demands without compromising model accuracy. This approach is particularly suitable for large-scale graphs, where computational efficiency is critical.

- **ACE-GLT** (Wang et al., 2023): Adaptive Compression and Efficient Gradient Learning Transmission (ACE-GLT) is a centralized sparsification method that adaptively compresses gradients during transmission. This technique effectively reduces communication costs while maintaining model performance, making it especially advantageous in environments with bandwidth constraints or high communication costs.

- **PruneFL** (Jiang et al., 2022): PruneFL employs an adaptive, distributed pruning strategy within federated learning to optimize performance on resource-constrained edge devices. Initially, an aggressive pruning phase reduces model size on a chosen client, followed by further pruning during training across clients. This iterative approach lowers both communication and computational demands while preserving model accuracy, making it highly suitable for deployment in edge environments with limited resources.

- **FedTiny** (Huang et al., 2023): FedTiny is a distributed pruning framework tailored to produce compact neural networks for federated learning on devices with limited memory and computational capacity. It employs an adaptive batch normalization selection to address data heterogeneity and a progressive pruning module that incrementally adjusts pruning policies at each layer. This design effectively minimizes computational and memory overhead while preserving model accuracy, making it well-suited for deployment in resource-constrained edge environments.

## C. Ablation with Different Number of Clients

In this section, we conduct ablation experiments on *the number of clients*. Figure 6a and Figure 6b respectively show the system performance for 10, 30, 50, 70, and 100 clients on Ogbn-Arxiv and Ogbn-Proteins. As the number of clients increases, overall performance tends to decline. However, EAGLES consistently outperforms the baseline in most cases. For instance, with 100 clients on Ogbn-Proteins, EAGLES surpasses the baseline by 2.13%, demonstrating its resilience. In contrast, on Ogbn-Arxiv with 100 clients, EAGLES shows a 0.26% decrease compared to the baseline. This may be attributed to the smaller size of Ogbn-Arxiv, where splitting the dataset among 100 clients results in fewer training nodes per client, limiting the model's ability to effectively learn patterns inherent in the dataset graph.

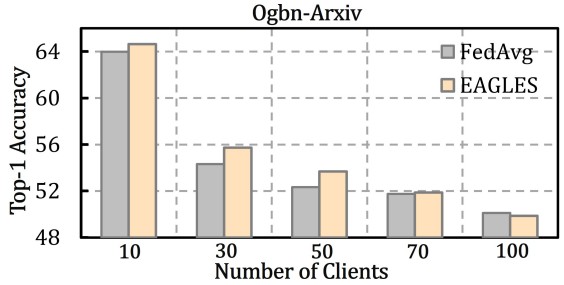
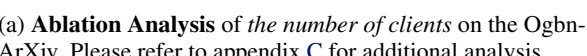
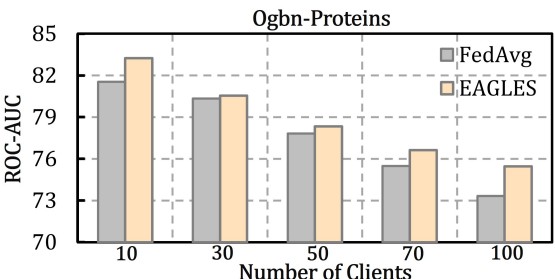

(a) **Ablation Analysis** of *the number of clients* on the Ogbn-ArXiv. Please refer to appendix C for additional analysis.

(b) **Ablation Analysis** of *the number of clients* on the Ogbn-Proteins. Please refer to appendix C for additional analysis.

## D. Sensitivity Analysis

In this section, we address **Q4** by conducting extensive ablation studies. Table 2 and Table 4 present a sensitivity analysis across various sparsification rates. **Obs. ❶** When fixing the row and observing the column data, we observe that overall model performance follows a trend of initially increasing and then decreasing, with optimal performance achieved when $S^{\mathcal{G}}$ is approximately 30%. **Obs. ❷** Fixing the column and analyzing the row data reveals that parameter sparsification can be effectively deepened, achieving optimal performance in the range of 50% to 70% for $S^{\mathbf{W}}$, although slight variations may arise depending on the dataset. **Obs. ❸** Simultaneous application of graph-level and parameter-level sparsification leads to significant improvements in model performance; for instance, in Ogbn-Arxiv with $(S^{\mathcal{G}}, S^{\mathbf{W}}) = (30\%, 70\%)$, accuracy increases by 1.38% ↑, while in Ogbn-Proteins with $(S^{\mathcal{G}}, S^{\mathbf{W}}) = (30\%, 50\%)$, accuracy improves by 1.70% ↑.

Figure 7a and Figure 7b illustrate the ablation analysis of two key parameters in our method: $\lambda_2$ and the number of sparsification experts $N^e$. **Obs. ❹** We observed that $\lambda_2$ exerts minimal influence on model performance, reaching its peak within the range of 0.3 to 0.4. **Obs. ❺** The classification accuracy exhibits an increasing trend with the growth of $N^e$. In our experiments, we set $N^e$ to 3, which prevents the introduction of excessive experts while still achieving relatively superior performance.

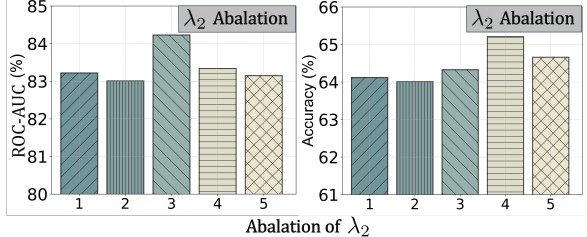
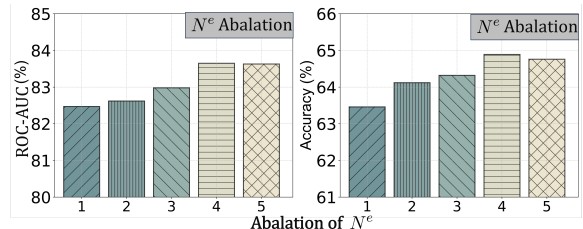

(a) **Ablation Analysis** of $\lambda_2$ on the Ogbn-Proteins and Ogbn-ArXiv. Please refer to appendix D for additional analysis.

(b) **Ablation Analysis** of the number of esparsification experts $N^e$ on the Ogbn-Proteins and Ogbn-ArXiv. Please refer to appendix D for additional analysis.

# E. Implementation Details

The experiments are conducted using NVIDIA GeForce RTX 3090 GPUs as the hardware platform, coupled with an Intel(R) Xeon(R) Gold 6240 CPU @ 2.60GHz. The deep learning framework employed is Pytorch, version 2.0.1, alongside CUDA version 11.7. For the small datasets Cora, pubmed and Ama-photo we utilized 2-layer GCN. For the larger datasets ogbn-arxiv and ogbn-products we employed 4-layer GraphSage and applied 4-layer DeeperGCN to ogbn-proteins. The default number of clients is set to 10.

# F. Quantitative analysis of heterogeneity reduction

We compute the overall similarity between two graphs by equally weighting structure similarity and feature similarity (50% each). The computation methods for each are as follows:

$$S_{\text{struct}} = 1 - \frac{|E_i - E_j|}{E_i + E_j + \epsilon}, \qquad S_{\text{feature}} = \frac{\overline{F}_i \cdot \overline{F}_j}{\|\overline{F}_i\| \|\overline{F}_j\|}, \qquad S_{\text{total}} = 0.5 \cdot S_{\text{struct}} + 0.5 \cdot S_{\text{feature}}$$

where $E_i$ and $E_j$ denote the number of edges in the two graphs, and $\epsilon$ is included to prevent division by zero. The average node feature vector for graph $i$ is defined as $\overline{F}_i = \frac{1}{|V_i|} \sum_{v \in V_i} F_v$, the feature vectors are truncated to the same dimension $d_{\min} = \min(d_i, d_j)$ during computation.

Employing the above calculation, we present heatmap visualizations for 10 clients on the PubMed and Ogbn-arxiv datasets. These visualizations illustrate the client subgraphs both at initialization and after five rounds of communication. Notably, an overall increase in isomorphism among clients can be observed, suggesting improved structural alignment through communication. The results are shown below.

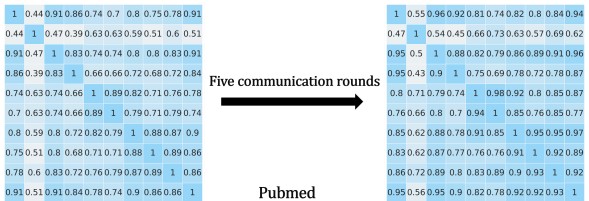

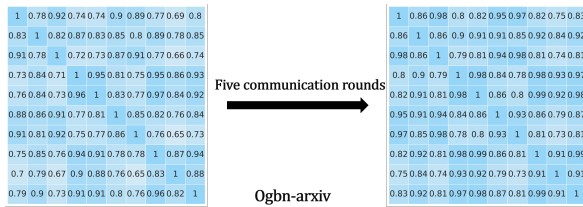

(a) **Quantitative analysis** of *heterogeneity reduction* on Pubmed.

(b) **Quantitative analysis** of *heterogeneity reduction* on Ogbn-arxiv.

