# OpenReview forum: "EAGLES: Towards Effective, Efficient, and Economical Federated Graph Learning via Unified Sparsification"
_ICML.cc/2025/Conference — ICML 2025 poster_

### Official Review · Reviewer_h7Za · 2025-03-03

**Overall Recommendation:** 4

**Summary:**

This paper introduces a unified framework that jointly considers graph-level and parameter-level sparsification. It incorporates dual experts and consensus-based sparsification to ensure a stable sparsification process. Extensive experiments demonstrate that the proposed method is effective, efficient, and economical.

**Claims And Evidence:**

The claims are supported by extensive experiments across datasets (Cora, Ogbn-Proteins) and metrics (FLOPS, ROC-AUC). Reductions in computational costs (82%↓ FLOPS) and communication (80%↓ bytes) are validated against baselines like FedAvg and ACE-GLT. However, claims about mitigating structural heterogeneity rely on qualitative arguments (e.g., "similar clients share knowledge via OT distance") without quantitative analysis of heterogeneity reduction.

**Essential References Not Discussed:**

The authors discuss and compare a wide range of related methods.

**Experimental Designs Or Analyses:**

Experiments are thorough, covering multiple datasets, sparsity levels, and baselines. Ablation studies (Table 2) validate parameter-graph sparsity interplay. However, the impact of expert count (Figure 7b) is under-discussed.

**Methods And Evaluation Criteria:**

The methods are well-suited for FGL challenges. Parameter sparsification avoids iterative pruning via dynamic masking, and graph sparsification addresses structural overfitting through multi-criteria experts. Evaluation on diverse datasets (small to large-scale) and metrics (FLOPS, ROC-AUC) is comprehensive.

**Other Comments Or Suggestions:**

The computational complexity of EAGLES could be better articulated, particularly regarding how its sparsification techniques scale with increasing data size or client count. This would provide a clearer view of the system’s scalability in large federated environments.

**Other Strengths And Weaknesses:**

Strengths:
(1) The paper effectively identifies a critical challenge in federated graph learning (FGL): the high computational cost and communication overhead when training GNNs on large-scale federated datasets. By introducing EAGLES, a unified sparsification approach, the authors provide a clear solution that addresses both graph and parameter sparsification, ensuring efficiency without sacrificing model performance.
(2) The extensive set of experiments conducted across various benchmark datasets, including ogbn-proteins and Pubmed, demonstrates the practical effectiveness of the proposed method. The substantial reductions in training FLOPS and communication costs, achieved while maintaining or even improving model accuracy, provide strong empirical evidence of the method’s efficiency and scalability.

Weaknesses:
(1)While the method demonstrates significant improvements in computational efficiency, a clear computational complexity analysis would help contextualize the performance gains.

**Questions For Authors:**

(1)Could the authors clarify how the Optimal Transport (OT) method adapts to the federated setting when client graphs have significant structural variations? Would this method still function efficiently if the number of clients increased substantially?
(2)How does $W_{gate}$ impact parameter sparsification when the number of GSEs increases? Would it lead to an excessive amount of additional parameters?

**Relation To Broader Scientific Literature:**

EAGLES makes a significant contribution by introducing a unified sparsification framework. The dual-expert approach, which builds upon MoE methods, adapts them for federated graph learning, an area that has seen limited exploration.

**Theoretical Claims:**

The manuscript’s mathematical formulation is generally free from notable errors; however, it lacks an analysis of computational complexity, which would provide a clearer understanding of the scalability and practical applicability of the proposed methods.

---

> ### Author Rebuttal · Authors · 2025-03-31
>
> # Dear Reviewer h7Za
>
> We sincerely thank you for your insightful feedback and have provided detailed responses to your questions.
>
> > ` W1: Without quantitative analysis of heterogeneity reduction.`
>
> We provide a quantitative analysis of heterogeneity reduction at [this link](https://anonymous.4open.science/r/Appendix-67CF/README.md).
>
> > `W2: Lack an analysis of computational complexity & S1: How its sparsification techniques scale with increasing data size or client count.`
>
> We analyzed the computational complexity from three aspects:
>
> 1. **Parameter Sparsification Module**:
>    - Forward/Backward FLOPs are $O(s_p \cdot d)$, where $s_p$ is the parameter sparsity rate and $d$ is the parameter dimension.
>    - Communication costs are reduced to $O(s_p \cdot d)$ via bit-wise mask compression (Section 4.2).
>    - Mask alignment requires $O(K \cdot L)$ operations per round, but with small constants for $K$ (clients) and $L$ (layers), its impact is negligible.
>
> 2. **Graph Sparsification Module**:
>    - With $T$ GSEs, the local computation is $O(T \cdot |E|)$, where $|E|$ is the number of edges.
>    - The message passing process has a complexity of $O(s_g \cdot |E| \cdot d)$, with $s_g$ as the graph sparsification rate.
>    - The gating mechanism adds $O(N \cdot D)$ operations, but since $D$ is typically small, its overhead is minimal.
>
> 3. **OT-based Similarity Computation**:
>    - Standard OT complexity is $O(n^3)$ for $n$-node graphs, but we reduce this to $O(n \log n)$ using the sliced Wasserstein distance.
>
> Overall, the computational complexity of EAGLES is given by:
> $$
> O\Big(s_p \cdot d + T \cdot |E| + s_g \cdot |E| \cdot d + n \log n\Big)
> $$
> Ignoring smaller constants, this simplifies to:
> $$
> O(d + |E| + n \log n)
> $$
> In summary, EAGLES scales linearly with the data size, and the number of clients has minimal impact on the computational complexity.
>
>
>
> > `W3: However, the impact of expert count (Figure 7b) is under-discussed.`
>
> As shown in Figure 7b (Appendix D.3), performance improves consistently as the number of experts increases, reaching its peak around 4 experts due to the benefit of richer structural perspectives. Beyond this point, the marginal gains diminish. While Section 5.4 briefly touches on this point, we agree that a more in-depth analysis would further strengthen the discussion.
>
>
> > `Q 1.1: Could the authors clarify how the OT method adapts to the federated setting when client graphs have significant structural variations?  `
>
> In FL settings with significant structural variations, our OT adaptation relies on two key mechanisms. First, the graph synergy expert encodes node contextual features in the $W_{\text{gate}}$ matrix, forming a structure-aware semantic space via hard concrete distribution sampling. This enables OT to assess similarity based on learned semantics instead of raw topology. Second, by treating each client’s structural distribution as a probability measure over this space, we derive client-specific transport plans and similarity weights (Eqs. 20 and 22), which automatically assign lower weights to structurally dissimilar clients. Importantly, only the compact $W_{\text{gate}}$ parameters are transmitted, preserving privacy while allowing the server to compute OT plans with $O(n \log n)$ complexity through entropic regularization.
>
>
>
> > `Q 1.2: Would this method still function efficiently if the number of clients increased substantially?`
>
> EAGLES remains efficient even as the number of clients grows significantly. Our framework reduces communication and computation through parameter sparsification (dynamic mask consensus) and OT-based similarity aggregation, minimizing redundant interactions. Experiments (Appendix Fig. 6a & 6b) show that scaling to 100 clients on Ogbn-Proteins and Cora results in only a minor performance drop, while achieving an 18% reduction in Training FLOPS and a 20% reduction in Communication Bytes compared to baselines.
>
>
>
> > `Q 2.1: How does $W_{gate}$ impact parameter sparsification when the number of GSEs increases?`
>
> $W_{\text{gate}}$ is a learnable gating parameter. Additional GSEs introduce richer structural diversity, and by integrating sparsified subgraphs obtained from multiple criteria through $W_{\text{gate}}$, the robustness of graph sparsification is enhanced. This reduction in structural redundancy allows for the allocation of different gradient update weights to model parameters during backpropagation, thereby influencing parameter sparsification.
>
>
>
> > `Q 2.2: Would $W_{gate}$ lead to an excessive amount of additional parameters?`
>
> The gating parameter matrix $W_{\text{gate}}$ (Eq. (12)) is designed as a lightweight mapping layer with low dimensionality. Specifically, its parameter size is $D \times T$, where $D$ is the input feature dimension and $T$ denotes the number of experts. Since the number of experts is typically a small constant, $W_{\text{gate}}$ scales linearly with $D$, thereby not introducing an excessive number of parameters.

---

> > ### Comment · Reviewer_h7Za · 2025-04-02
> >
> > I have carefully reviewed the rebuttal and also checked the feedback from other reviewers. The authors' further explanation of computational complexity is convincing, and my questions have been well addressed. The work may have a potential impact and will accordingly increase my score.

---

> > > ### Author Response · Authors · 2025-04-02
> > >
> > > ### Dear reviewer h7Za
> > > Thank you for your thoughtful feedback and for reconsidering our work. Your comments helped us refine the presentation and strengthen the manuscript. We truly appreciate the opportunity to clarify our approach and the time you spent reviewing our submission.
> > >
> > >
> > > Best regards,
> > >
> > > Authors

---

### Official Review · Reviewer_Fguv · 2025-03-10

**Overall Recommendation:** 4

**Summary:**

The paper introduces EAGLES, a unified sparsification framework designed to enhance FGL by addressing computational and communication challenges. EAGLES optimizes both graph structures and model parameters through client-consensus parameter sparsification, which generates multiple unbiased subnetworks at various sparsity levels. The method also employs a dual-expert approach with graph sparsification and synergy experts, which improve the efficiency of message passing and reduce data overfitting. The comprehensive experimental results validate the effectiveness of the proposed method.

**Claims And Evidence:**

The paper provides a relatively clear explanation of its claims. FGL faces significant computational challenges when handling large-scale graph data. Figure 1 effectively illustrates this phenomenon. However, additional empirical studies could further corroborate this analysis and strengthen the claims presented in the paper.

**Essential References Not Discussed:**

The key contribution of the paper is the unified sparsification approach for FGL, but it only references a graph sparsification technique, DSpar, that sparsifies graph structures based on node degree. However, there is also a relevant method, DropEdge, introduced by [3], which applies random edge dropout to improve deep graph convolutional networks for node classification. This technique is particularly important for reducing computational costs while preserving graph structure and can be considered an essential reference for addressing graph sparsification challenges in the context of FGL, especially in comparison to the single-criterion sparsification discussed in the paper.

[3] Rong Y, Huang W, Zhang Y, et al. DropEdge: Towards Deep Graph Convolutional Networks on Graphs with Sparse Edge Features. arXiv preprint arXiv:2006.10616, 2020.

**Experimental Designs Or Analyses:**

Extensive experiments across six datasets and multiple backbones (GCN, GraphSAGE, DeeperGCN) strengthen validity. Ablation studies on sparsity rates and client numbers (Figures 4–7) convincingly demonstrate resilience.

**Methods And Evaluation Criteria:**

The proposed methodology and evaluation criteria align well with the problem of optimizing federated graph learning. The dual-expert sparsification approach appears to be a reasonable solution, and the chosen evaluation metrics (FLOPS and communication costs) are directly applicable to the problem at hand.

**Other Comments Or Suggestions:**

The manuscript specifies the split ratios for each dataset but does not describe the splitting strategy. The authors should include details on the splitting approach in the manuscript.

**Other Strengths And Weaknesses:**

**Strengths:**
- This paper introduces the first unified framework for both graph and parameter sparsification in FGL.
- The motivation behind this paper is explained with great clarity.
- This paper presents a novel use of Optimal Transport (OT) to measure client similarity, which is an interesting approach.

**Weaknesses:**
- There is a typo on page seven in Section 5 where "comprehensively" is misspelled as "omprehensively."
- Experiments focus primarily on academic citation and biological networks. There is no validation on social network graphs. Including relevant experiments would strengthen the generalizability and applicability of the proposed method.

**Questions For Authors:**

1.How does the proposed method perform in scenarios where clients have vastly different computational capabilities (e.g., edge devices versus more powerful systems)?

2.In the code, the authors only perform data partitioning using the Louvain method. Can the proposed approach still be effective under other non-iid partitioning methods, such as Metis?

**Relation To Broader Scientific Literature:**

EAGLES builds on federated learning (FedAvg, FedProx) and graph sparsification (DSpar [1]). The integration of MoE for graph pruning is novel, advancing prior work on MoE [2].

[1] Liu Z, Zhou K, Jiang Z, et al. DSpar: An Embarrassingly Simple Strategy for Efficient GNN Training and Inference via Degree-based Sparsification. arXiv preprint arXiv:2307.02947, 2023.

[2] Shazeer N, Mnih A, Ranzato M, et al. Outrageously large neural networks: The sparsely gated mixture-of-experts layer. arXiv preprint arXiv:1701.06538, 2017.

**Theoretical Claims:**

The theoretical section of the manuscript is relatively detailed. In particular, the Harmony Sparsification Principle and its impact on federated graph learning are interesting and well-reasoned, providing concrete theoretical guidance and practical reference for the design of sparsification frameworks.

---

> ### Author Rebuttal · Authors · 2025-03-31
>
> # Dear Reviewer Fguv
> We sincerely thank you for taking the time to evaluate our work and have adressed your concerns as follows:
>
> >  ` W1: Additional empirical studies addressing the significant computational challenges faced by FGL will further strengthen this analysis.`
>
> In the theoretical model, the message passing mechanism in GNNs causes the neighborhood size to expand exponentially with the number of layers. For a graph with an average degree of $d$, 1-hop neighborhood covers $d$ neighbors, a 2-hop neighborhood covers $d^2$ neighbors, and an $L$-hop neighborhood covers $d^L$ neighbors [1].
>
> We measured the k-hop receptive fields (k=1,2,3,4) for the amz-photo and Ogbn-arxiv datasets. The results are as follows:
>
> |   datasets    | 1-hop | 2-hop  |  3-hop  |  4-hop  |
> | :-----------: | :---: | :----: | :-----: | :-----: |
> | **amz-photo** | 32.13 | 802.86 | 2519.35 | 4681.62 |
>
> The results show that the receptive fields exhibit a clearly super-linear growth trend, confirming that GNNs indeed face significant computational challenges.
>
> [1]: Xu, K.; Hu, W.; Leskovec, J.; and Jegelka, S. (2019). How Powerful are Graph Neural Networks? arXiv preprint arXiv:1810.00826.
>
>
>
> > `W2: Supplementary experiments addressing the omitted DropEdge.`
>
> We conducted experiments on DropEdge on the PubMed, with some of the results presented below (we adopted the original 0.8 retention rate for GCN).
>
> **Pubmed**
>
> |   Methods    | Top-1 Accuracy | Max Training FLOPS | Communication BYTES |
> | :----------: | :------------: | :----------------: | :-----------------: |
> |  **FedAvg**  |     85.65      |    1x (2.49E9)     |     1x (6.19E9)     |
> | **DropEdge** |     85.78      |   0.90x (↓0.10x)   |   1.00x (↓0.00x)    |
> |  **EAGLES**  |   **86.97**    | **0.48x (↓0.52x)** | **0.37x (↓0.63x)**  |
>
> > `W3: A typo on page seven in Section 5 where "comprehensively" is misspelled as "omprehensively."`
>
> Thank you for your careful reading. We have corrected the typo and carefully proofread the manuscript to fix similar issues.
>
>
>
> > `W4: Validate the proposed method on social network graph datasets. `
>
> We conducted experiments on the Flickr dataset to validate the effectiveness of the proposed method on social network graph datasets:
>
> **Flickr**
>
> |   Methods   |  Top-1 Accuracy   | Max Training FLOPS | Communication BYTES |
> | :---------: | :---------------: | :----------------: | :-----------------: |
> | **FedAvg**  |       50.15       |    1x (8.49E9)     |     1x (4.67E9)     |
> |  **FGGP**   |   49.78(↓0.37)    |   1.23x (↑0.23x)   |   1.33x (↑0.33x)    |
> | **PruneFL** |   47.45(↓2.70)    |   0.77x (↓0.23x)   |   1.00x (↓0.00x)    |
> | **FedDIP**  |   50.33(↑0.17)    |   0.67x (↓0.33x)   |   0.83x (↓0.17x)    |
> | **EAGLES**  | **50.89** (↑0.74) | **0.48x (↓0.52x)** | **0.37x (↓0.63x)**  |
>
>
>
> > `S1: The manuscript specifies the split ratios for each dataset but does not describe the splitting strategy. `
>
> Thank you for pointing that out. We will include additional details on the splitting strategy in the revised manuscript.
>
> > `Q1: How does the proposed method perform in scenarios where clients have vastly different computational capabilities (e.g., edge devices versus more powerful systems)?`
>
> Clients with limited resources can opt for higher sparsity to reduce memory and computation, while more capable machines may choose lower sparsity for better accuracy. Additionally, consensus-based parameter masks and a multi-expert graph sparsification framework ensure that all clients benefit from an efficient, robust model.
>
>
>
> > `Q2: Can the proposed approach still be effective under other non-iid partitioning methods, such as Metis?`
>
> We conducted experiments on Cora and ogbn-arxiv, and the results are shown below:
>
> **Cora**
>
> |   Methods   |  Top-1 Accuracy   | Max Training FLOPS | Communication BYTES |
> | :---------: | :---------------: | :----------------: | :-----------------: |
> | **FedAvg**  |       70.62      |    1x (6.72E8)     |     1x (6.02E9)     |
> |  **FGGP**   |   69.58 (↓1.04)   |   1.42x (↑0.42x)   |   1.18x (↑0.18x)    |
> | **PruneFL** |   67.94 (↓2.68)   |   0.57x (↓0.43x)   |   1.00x (↓0.00x)    |
> | **FedDIP**  |   70.38 (↓0.24)   |   0.61x (↓0.39x)   |   0.59x (↓0.41x)    |
> | **EAGLES**  | **71.27** (↑0.65) | **0.48x (↓0.52x)** | **0.37x (↓0.63x)**  |
>
> **Ogbn-arxiv**
>
> |   Methods   |    Top-1 Accuracy     | Max Training FLOPS | Communication BYTES |
> | :---------: | :-------------------: | :----------------: | :-----------------: |
> | **FedAvg**  |         55.30         |    1x (1.58E10)    |     1x (6.14E9)     |
> |  **FGGP**   |     55.09(↓0.21)      |   5.66x (↑4.66x)   |   1.45x (↑0.45x)    |
> | **PruneFL** |     52.34 (↓2.96)     |   0.69x (↓0.31x)   |   1.00x (↓0.00x)    |
> | **FedDIP**  |     55.32 (↑0.02)     |   0.48x (↓0.52x)    |   0.59x (↓0.41x)    |
> | **EAGLES**  | **56.89** **(↑1.59)** | **0.35x (↓0.65x)** | **0.47x (↓0.53x)**  |
>
> The results show that under Metis partitioning, EAGLES still exhibits superior performance.

---

### Official Review · Reviewer_jagF · 2025-03-13

**Overall Recommendation:** 4

**Summary:**

EAGLES introduces a framework designed to reduce computational and communication costs in federated graph learning by jointly sparsifying both model parameters and graph structures. It employs client-consensus pruning to generate subnetworks at different sparsity levels and utilizes a mixture of experts for graph sparsification. This approach achieve substantial reductions in FLOPs and communication overhead across various datasets, all while preserving accuracy. Results show improved performance over baselines in node classification tasks.

**Claims And Evidence:**

The claims are largely supported by experiments across six datasets (Cora, Pubmed, OGB benchmarks) and comparisons with 14 baselines. Evidence includes:

1. Table 1 shows EAGLES outperforms FedAvg/FedProx in accuracy (e.g., +1.32% on Pubmed) while reducing FLOPs (52%) and communication (63%).

2. Ablation studies (Table 2, Figure 5) validate the impact of sparsity levels.

3. Theoretical grounding via the Harmony Sparsification Principle (Eq. 3) aligns with empirical results.

**Essential References Not Discussed:**

This manuscript compares and discusses quite a few baseline methods.

**Experimental Designs Or Analyses:**

The framework is validated across 6 datasets with diverse backbones (GCN, GraphSAGE) and compared against 14 baselines, including the state-of-the-art federated graph learning method (FedTAD) and pruning approaches like ACE-GLT.

**Methods And Evaluation Criteria:**

1. Parameter Sparsification: Dynamic threshold optimization with STE and consensus masking (Eq. 4-7) is novel and suitable for federated settings.

2. Graph Sparsification: Dual experts (GSE/GSyE) with hard concrete distribution (Eq. 13-17) effectively address structural heterogeneity.
Evaluation：

3. Metrics (Top-1 Accuracy, ROC-AUC, FLOPs, communication bytes) are standard and comprehensive.

**Other Comments Or Suggestions:**

It is suggested to supplement experiments on training time to further verify the efficiency of the method.

**Other Strengths And Weaknesses:**

Strengths:

1. The paper creatively bridges federated learning and graph sparsification, addressing both computational and structural challenges in FGL. This dual focus (parameter + graph sparsification) is novel and addresses a critical gap in federated graph learning literature.

2. The framework’s ability to handle large-scale graphs (e.g., ogbn-proteins with 132,534 nodes) demonstrates real-world applicability.

Weaknesses:

1. Experiments focus on homophilic graphs (e.g., Cora, OGB). Performance on heterophilic graphs (e.g., arXiv) remains unvalidated.

2. Whether the proposed method can speed up the experiments was not explored.

**Questions For Authors:**

Can the authors provide additional ablation experiments regarding $\lambda_2$ and $\lambda_3$ in Eq (19)?

**Relation To Broader Scientific Literature:**

The contributions of the paper are related to the broader scientific literature of following areas.

1. FGL: Improves FedAvg/FedProx by addressing graph/parameter redundancy.

2. MoE: Adapts mixture-of-experts to graph sparsification (novel).

3. Pruning: Unifies model/graph pruning, unlike DSpar/ACE-GLT.

**Theoretical Claims:**

The theoretical claims are basically correct. However, what $W_{gate}$ refers to in eq (12) lacks the necessary explanation in the text.

---

> ### Author Rebuttal · Authors · 2025-03-31
>
> # Dear Reviewer jagF
>
> We sincerely appreciate your detailed review and invaluable feedback. In the response below, we provide a thorough reply to address your concerns and offer a clearer explanation of our method.
>
> > ` W1: what $W_{gate}$ refers to in eq (12) lacks the necessary explanation in the text.`
>
> $W\_{\text{gate}}$ is a learnable weight matrix in the Graph Synergy Expert (GSyE) module that performs gating. It projects the node feature matrix $X$ into a latent space for Hard Concrete sampling, determining which graph sparsification experts to activate. In essence, $W\_{\text{gate}}$ generates gating vectors $z$ that, after thresholding, indicate the significance of each edge in the final sparsified subgraph. We will introduce $W_{\text{gate}}$ in the revised manuscript.
>
>
>
> > ` W2: Performance on heterophilic graphs (e.g., arXiv) remains unvalidated.`
>
> We conducted experiments on heterophilic graphs using ogbn-arxiv-TA [1] from the HeTGB (Heterophilic Text-attributed Graph Benchmark):
>
> |   Methods   | Top-1 Accuracy |   Max Training FLOPS    |     Communication      |
> | :---------: | :------------: | :---------------------: | :--------------------: |
> | **FedAvg**  |     64.24      |      1x (1.78E10)       |      1x (6.16E9)       |
> | **PruneFL** |     61.37      |          0.58x          |         1.00x          |
> | **FedTAD**  |     64.92      |         32.42x          |         1.11x          |
> | **EAGLES**  |   **65.89**    | **0.33x  (↓0.68x**) | **0.48x** (↓**0.52x**) |
>
> The results show that on heterophilic graphs using ogbn-arxiv-TA, EAGLES also demonstrates superiority.
>
> [1]: Li, S.; Wu, Y.; Shi, C.; and Fang, Y. (2025). *HeTGB: A Comprehensive Benchmark for Heterophilic Text-Attributed Graphs*. arXiv preprint arXiv:2503.04822.
>
>
>
> > ` W3 & S1: Whether the proposed method can speed up the experiments was not explored, and supplement experiments on training time to further verify the efficiency of the method.`
>
> We measured the time required for clients to reach the target accuracy on the ogbn-arxiv dataset across different methods, and the accuracy achieved by different methods at the same number of epochs on the ogbn-proteins dataset. The results are presented below:
>
> **TIME TO REACH TARGET ACCURACY**
>
> |   Methods   | Time to reach 70% accuracy | Time to reach 80% accuracy | Time to reach 90% accuracy |
> | :---------: | :------------------------: | :------------------------: | :------------------------: |
> | **FedAvg**  |          54.23 s           |          223.42 s          |          376.23 s          |
> | **FedTiny** |          35.28 s           |          149.08 s          |          281.23 s          |
> | **FedDIP**  |          36.44 s           |          155.68 s          |          278.62 s          |
> | **PruneFL** |          40.52 s           |          177.21 s          |          319.53 s          |
> | **EAGLES**  |        **14.04 s**         |        **85.94 s**         |        **162.79 s**        |
>
> **ROC-AUC UNDER THE SAME EPOCH**
>
> |   Methods   | EPOCH: 50 | EPOCH: 150 | EPOCH: 300 |
> | :---------: | :-------: | :--------: | :--------: |
> | **FedAvg**  |  70.34   |   80.38    |   81.49    |
> | **FedTiny** |   71.36   |   77.68    |   79.90    |
> | **FedDIP**  |   70.98   |   78.59    |   81.33    |
> | **PruneFL** |   69.22   |   76.23    |   78.69    |
> | **EAGLES**  | **73.97** | **81.85**  | **82.32**  |
>
> The results show that EAGLES can train the model to a target accuracy in a shorter amount of time, and also achieve higher performance at the same number of epochs. This further verifies that our proposed method can accelerate model training.
>
>
>
> > `Q1: Can the authors provide additional ablation experiments regarding $λ_2$ and $λ_3$ in Eq(19)`
>
> We conducted ablation experiments on $λ_2$ and $λ_3$ on the ogbn-arxiv dataset, and the results are presented below:
>
> **Ogbn-arxiv** (fix $λ_3$ = 1e-6)
>
> |    dataset     | $λ_2$=0.1 |    $λ_2$=0.05    |    $λ_2$=0.2     |
> | :------------: | :----: | :-----------: | :-----------: |
> |   **Pubmed**   | 86.97  |     86.23 (↓0.74)     | 86.96 (↓0.01) |
> |   **photo**    | 92.31  | 92.75 (↑0.44) | 91.85 (↓0.46) |
> | **Ogbn-arxiv** | 65.37  | 64.92 (↓0.45) | 65.33 (↓0.04) |
>
> **Ogbn-arxiv** (fix $λ_2$ = 0.1)
>
> |    dataset     | $λ_3$=1e-6 |    $λ_3$=1e-5    |    $λ_3$=1e-7    |
> | :------------: | :-----: | :-----------: | :-----------: |
> |   **Pubmed**   |  86.97  | 84.22 (↓2.75) | 86.89 (↓0.08) |
> |   **photo**    |  92.31  | 90.56 (↓1.75) | 92.15 (↓0.16) |
> | **Ogbn-arxiv** |  65.37  | 64.38 (↓0.99) | 65.42 (↑0.05) |
>
> Increasing $λ_2$ enforces stricter GSyE constraints, balancing expert contributions and enhancing subgraph homogeneity. In heterogeneous scenarios, raising $λ_2$ from 0.05 to 0.2 leads to a clear accuracy boost. Conversely, while a larger $λ_3$ speeds up sparse parameter identification, it may cause significant accuracy loss; a smaller $λ_3$ permits finer-grained sparsification.

---

> > ### Comment · Reviewer_jagF · 2025-04-06
> >
> > I have read the rebuttal and appreciate the authors' response. The additional experiments further validate the effectiveness of the proposed method. A minor suggestion is to include the explanation for weakness 2 in the paper if it has not already been added. I maintain my score and support the acceptance of the paper.

---

> > > ### Author Response · Authors · 2025-04-06
> > >
> > > ### Dear Reviewer jagF,
> > >
> > > We sincerely appreciate your invaluable support for our research. Your insightful suggestions regarding the scalability and flexibility of EAGLES have significantly contributed to improving the depth and precision of our manuscript. It has been an honor to incorporate your comments and strengthen our work accordingly. Thank you once again for your time, expertise, and constructive review.
> > >
> > > Best regards,
> > >
> > > Authors

---

### Official Review · Reviewer_PxDJ · 2025-03-15

**Overall Recommendation:** 2

**Summary:**

This work introduces EAGLES, a framework for Federated Graph Learning (FGL) that reduces computational demands while maintaining high performance. By unifying graph and model sparsification, it simplifies graph structures and prunes model parameters efficiently. EAGLES uses multi-criteria experts to sparsify graphs and integrates results using a synergy expert, ensuring better knowledge sharing across clients with diverse data.

**Claims And Evidence:**

Yes

**Essential References Not Discussed:**

Yes

**Experimental Designs Or Analyses:**

Yes

**Methods And Evaluation Criteria:**

Yes

**Other Comments Or Suggestions:**

N.A

**Other Strengths And Weaknesses:**

Strengths:

- EAGLES introduces a unified sparsification framework that simultaneously sparsifies graph structures and model parameters. It uses multi-criteria graph sparsification experts and a synergy expert to reduce graph size while preserving critical structural information.

- By addressing key challenges like data heterogeneity, computational inefficiency, and communication overhead, EAGLES provides a scalable and economical solution for Federated Graph Learning. Consensus-based parameter sparsification further ensures efficient pruning without iterative adjustments, addressing computational and communication overhead.

- The evaluation shows resilience under varying sparsification rates and client distributions, making it effective for large-scale federated graph learning.

Weaknesses:

- While the paper introduces the Graph Synergy Expert to integrate sparsified subgraphs from multiple experts, it would benefit from a more detailed step-by-step explanation of how the GSyE processes and combines the outputs of various sparsification experts.

For instance, describing how the hard concrete distribution is optimized and how the gating mechanism selects and integrates key structural information for each node would enhance clarity.

- The consensus-based parameter sparsification strategy is described briefly, but its implementation could use more depth. A detailed breakdown of how the dynamic masking thresholds are computed, how client-specific masks are aligned, and how the rollback pruning strategy ensures pruning stability would be valuable.

- Elaborating on how the trade-offs between structural similarity, computational efficiency, and model performance are balanced in the optimization process (Equation 3) would strengthen the framework.

**Questions For Authors:**

The paper introduces the Graph Synergy Expert (GSyE) to integrate sparsified subgraphs from multiple experts, but a more detailed, step-by-step explanation of its process would enhance clarity. Specifically, describing how the GSyE optimizes the hard concrete distribution and how the gating mechanism selects and integrates key structural information for each node would provide valuable insights. Including examples, pseudocode, or visualizations could further illustrate the functionality and importance of this component.

The consensus-based parameter sparsification strategy is briefly mentioned, but its implementation could benefit from greater depth. A detailed explanation of how dynamic masking thresholds are computed, how client-specific masks are aligned, and how the rollback pruning strategy ensures pruning stability would make the approach more comprehensible.

Additionally, elaborating on how the framework balances trade-offs between structural similarity, computational efficiency, and model performance in the optimization process (Equation 3) would offer a stronger understanding of its effectiveness.

**Relation To Broader Scientific Literature:**

Yes

**Theoretical Claims:**

Yes

---

> ### Author Rebuttal · Authors · 2025-03-31
>
> We sincerely appreciate for taking the time to review our manuscript and hope our response will address your concerns and contribute to an improved score.
>
> > ` W1: How the GSyE processes and combines the outputs of various sparsification experts.`
>
> In our method, GSyE (Graph Synergy Expert) is used to fuse and refine different subgraphs generated by multiple GSE (Graph Sparsification Expert, aiming to alleviate *structure information overfitting* during the graph sparsification process. The specific steps are as follows:
>
> 1. Multiple GSE generate various versions of subgraphs based on different criteria (Eq. (11)).
> 2. For the target node *v*, its node features $\mathbf{X}$ are projected using a learnable matrix $\mathbf{W}_{gate}$ to obtain gating scores $\boldsymbol{z}$, which are then used to generate continuous approximations $\psi(\boldsymbol{z})$ of binary gates, thereby enabling gradient-based optimization (Eq. (13)).
> 3. The HardStep function (Eq. (14)) is applied to hard-threshold $\psi(\boldsymbol{z})$ to either 0 or 1, determining whether to activate the corresponding expert's recommendation for the edge $e_{ij}$.
> 4. For each edge $e_{ij}$, if at least one expert recommends retaining it, the edge is marked as a candidate edge; otherwise, it is pruned directly.
>
>
>
> > `Q1: How the hard concrete distribution is optimized?`
>
> The hard concrete distribution is applied to the raw gating scores $\boldsymbol{z}$ (Eq. (12)) to produce continuous probabilities $\psi(\boldsymbol{z})$. The HardStep function further binarizes $\psi(\boldsymbol{z})$ into discrete gates—0 and 1. During backpropagation, we employ the Straight-Through estimator [1] to approximate gradients and address the optimization problem.
>
> [1]: Bengio, Y.; Léonard, N.; and Courville, A. (2013). *Estimating or propagating gradients through stochastic neurons for conditional computation*. arXiv preprint arXiv:1308.3432.
>
>
>
> > `Q2: How dynamic masking thresholds are computed? + Q3: How client-specific masks are aligned?`
>
> In consensus-informed parameter sparsification, dynamic masking thresholds are computed through a layer-wise adaptive process. For the $l$-th layer’s parameter matrix $W^{(l)}$, a threshold vector $\kappa\_{0}^{(l)}$ is dynamically optimized using straight-through estimators (STE) to bypass non-differentiability during backpropagation. Specifically, parameters $W^{(l)}$ are pruned if their absolute values fall below $\kappa\_{0}^{(l)}$, where thresholds are updated via a loss function (Eq. (6)) to maximize sparsity while maintaining performance. Client-specific masks are aligned through a consensus mechanism where overlapping “1”s in binary masks across clients form a unified sparse subnetwork, enabling parameter sharing and communication efficiency.
>
>
>
> > `Q4: How the rollback pruning strategy ensures pruning stability?`
>
> The rollback pruning strategy ensures pruning stability by designating, at each predefined pruning checkpoint (for example, every 10% increment in pruning rate), the highest-performing subnetwork within an acceptable accuracy range (±3%). Before moving on to a deeper level of pruning, the method reverts (rolls back) to this optimal subnetwork. This rollback step prevents the accumulation of errors from continuous pruning and mitigates sudden drops in performance, thereby maintaining overall model stability and ensuring effective deep pruning.
>
>
>
> > `Q5: How the framework balances trade-offs between structural similarity, computational efficiency, and model performance in the optimization process (Equation 3)?`
>
> Our weighted loss function combines a structural similarity term (enforcing alignment of sparse subnetworks via mask consensus), a computational cost penalty (promoting parameter sparsity to reduce FLOPs), and a task-specific performance loss (e.g., cross-entropy for accuracy). Hyperparameters $λ\_1$ and $λ\_2$ dynamically adjust the trade-offs: higher $λ\_1$ prioritizes mask consistency across clients, while $λ\_2$ controls sparsity-intensity.

---

### Decision · Program_Chairs · 2025-05-01

**Decision:**

Accept (poster)

**Comment:**

Three reviewers gave it a score of 4. The paper makes a clear and compelling case for the need to jointly consider parameter-level and data-level sparsification in FGL systems, a perspective that is both original and timely given the increasing deployment of GNNs in resource-constrained federated environments. The paper is backed by strong motivation and a novel problem setting and the authors provided additional analysis on computational complexity and clarified certain architectural details.Therefore, this paper can make it a valid contribution to ICML, and the decision is to recommend Acceptance.